# communications
# earth & environment

# Viruses of sulfur oxidizing phototrophs encode genes for pigment, carbon, and sulfur metabolisms

Poppy J. Hesketh-Best [1,5], Alice Bosco-Santos[2,5], Sofia L. Garcia[1], Molly D. O'Beirne[3], Josef P. Werne [3], William P. Gilhooly III [4] & Cynthia B. Silveira [1✉]

Viral infections modulate bacterial metabolism and ecology. Here, we investigated the hypothesis that viruses influence the ecology of purple and green sulfur bacteria in anoxic and sulfidic lakes, analogs of euxinic oceans in the geologic past. By screening metagenomes from lake sediments and water column, in addition to publicly-available genomes of cultured purple and green sulfur bacteria, we identified almost 300 high and medium-quality viral genomes. Viruses carrying the gene *psbA*, encoding the small subunit of photosystem II protein D1, were ubiquitous, suggesting viral interference with the light reactions of sulfur oxidizing autotrophs. Viruses predicted to infect these autotrophs also encoded auxiliary metabolic genes for reductive sulfur assimilation as cysteine, pigment production, and carbon fixation. These observations show that viruses have the genomic potential to modulate the production of metabolic markers of phototrophic sulfur bacteria that are used to identify photic zone euxinia in the geologic past.

---

[1] Department of Biology, University of Miami, Coral Gables, FL, USA. [2] Institute of Earth Surface Dynamics, University of Lausanne, Lausanne, Switzerland. [3] Department of Geology & Environmental Science, University of Pittsburgh, Pittsburgh, PA, USA. [4] Department of Earth Sciences, Indiana University-Purdue University Indianapolis, Indianapolis, IN, USA. [5] These authors contributed equally: Poppy J. Hesketh-Best, Alice Bosco-Santos. ✉email: cynthiasilveira@miami.edu

In the Archean (from ~4.0 to 2.5 billion years ago) and Proterozoic (from ~2.5 to 0.5 billion years ago) eons, prior to the development of present oxygen levels in the oceans and atmosphere, anoxygenic photosynthetic bacteria may have been major marine primary producers[1–5]. These phototrophs fix carbon under anaerobic conditions using inorganic electron donors, such as ferrous iron and hydrogen sulfide. In the Archean, anoxygenic photosynthetic ferrous iron-oxidizing bacteria could have sustained up to 10% of modern-day primary productivity[6–9]. In the Proterozoic, shallow and intermediate depths along continental margins experienced the expansion of oceanic euxinia[10], where sulfide oxidizers catalyzed most primary production and influenced the planet's oxidant balance[1–3,11]. The presence of biomarkers suggests that sulfide-oxidizing phototrophs were also common during oceanic anoxia events linked to mass extinctions throughout the Phanerozoic[12,13].

Modern euxinic lakes hosting sulfide oxidizing phototrophs provide a unique opportunity for identifying biosignatures of relict oceans potentially preserved in the geologic record. These anoxic phototrophs are green sulfur bacteria (GSB, family Chlorobiaceae) and purple sulfur bacteria (PSB, families Chromatiaceae and Ectothiorhodospiraceae) that inhabit the photic zone euxinia, where sulfide reaches the sunlit portions of stratified anoxic water columns[14,15]. These primary producers have narrow optimal requirements of micro-oxic to anoxic conditions, free sulfide, and sunlight. PSB are more tolerant to dissolved oxygen and GSB are adapted to lower light levels, with a particular brown-pigmented group having even lower light requirements[16]. Consequently, GSB and PSB light-harvesting pigments and their diagenetic products preserved in the geologic record represent biomarkers that provide clues about past biological processes and environmental conditions[17–19]. Based on the ecology of modern euxinic basins, the preservation of diagenetic products of GSB carotenoid pigments (chlorobactene and isorenieratene, preserved as chlorobactene and isorenieratane, respectively) is interpreted as a marker for a deeper photic zone, compared to where PSB pigments (okenone, preserved as okenane) are found[18,20]. Yet, a growing body of evidence shows that the distribution of GSB and PSB in modern euxinic water columns is not as tightly correlated to physical and chemical conditions (oxygen, sulfide, and light) as previously thought. In the euxinic Green Lake (NY), okenone is the major biomarker of sulfide oxidizers in sediments, while GSB is dominant in the water column[21]. Additionally, the amount of okenone observed in pure cultures of PSB is decoupled from cell densities and suggests that the expression of this pigment is inducible[22]. These observations imply that okenone concentrations in sediments depend on metabolic rates and not solely on PSB abundance.

Long-term studies of euxinic Lake Cadagno, Switzerland, further show a decoupling between the abundance of sulfide oxidizing phototrophs and carbon fixation rates. In a growing season, one species of PSB, *Chromatium okenii*, accounted for only 0.3% of the bacterial community, and yet, it was responsible for 70% of the carbon uptake[23]. In subsequent growing seasons, GSB was dominant, representing 95% of the community, but the PSB *Thiodictyon syntrophicum* was responsible for 25.9% of the total carbon fixation[24]. Microbial sulfur cycling is also convoluted, as observed in Mahoney Lake[25]. There, the peak activity of PSB does not correspond to the peak supply of microbial sulfide production[16,25]. All these observations suggest that unknown biological interactions play an important role in defining the distribution of these phototrophs and their biogeochemical signals[16,26,27]. Here, we propose that a largely unexplored biotic factor controls the distribution and activity of anoxygenic sulfide oxidizing phototrophs: viral infection.

Bacteriophages, also known as phages, are viruses that infect bacteria and can laterally transfer genes, modulate gene expression, and control host population dynamics[28–31]. In the modern surface ocean, viral predation is responsible for the daily turnover of about 25% of the bacterioplankton[32]. Phages infecting oxygenic phototrophs (Cyanobacteria) encode many genes involved in the synthesis of light-harvesting pigments (*ho1*, *pebS*, *cpeT*, *pcyA*)[32], which have been experimentally demonstrated to alter photosynthetic rates[33]. Cyanophages also encode genes for enzymes that block carbon fixation through the Calvin Cycle during infection while increasing nucleotide production through the Pentose Phosphate Pathway[34]. Most of these carbon metabolism pathways, as well as nucleotide and protein synthesis pathways, are shared between Cyanobacteria and sulfide oxidizing phototrophs[35]. These observations lead to the hypothesis that phage infections could play a role in GSB and PSB ecology and the biogeochemical cycles they modulate in euxinic lakes. A recent study showed that lake GSB populations were simultaneously infected with 2–8 viruses per cell[36]. One GSB host was consistently associated with two prophages with a nearly 100% infection rate for over 10 years[36]. High rates of horizontal gene transfer are also suggested in GSB genomic signatures, reaching 24% of all genes in *Chlorobaculum tepidum*[37]. If these frequent phage infections modify the genomes and physiology of these primary producers, the implications could extend to biosignatures in the rock record. For example, phage regulation of phototrophic sulfur bacteria pigment synthesis may affect the abundance and distribution of GSB and PSB biomarkers that are used as indicators of photic zone euxinia in the rock record.

Here, we identify through long-read metagenomic sequencing the genomes of phages putatively infecting GSB and PSB inhabiting euxinic lakes (Figs. 1a, b, and Supplementary Fig. 1). We combine these analyses with the identification of integrated phages in publicly available GSB and PSB genomes. The phage genomes identified here encode genes involved in pigment production, carbon fixation, and sulfur metabolism. These results show that GSB and PSB viruses have the genomic potential to manipulate hosts' biosignatures.

## Results

**Bacterial community composition.** Nanopore sequencing generated $3.9 \times 10^6$ reads from Lime Blue sediment and $19.2 \times 10^6$ reads from Poison Lake water (Supplementary Table 1 and Supplementary Fig. 2). Trimming and quality filtering removed 96 and 93% of reads from Lime Blue and Poison Lake, respectively. Assemblies generated 40,807 contigs from Lime Blue sediment metagenomes and 4310 from Poison Lake water metagenomes. Lime Blue and Poison Lake were dominated by members of the phylum Proteobacteria (48.77% of Lime Blue reads and 70.51% of Poison Lake reads; 45.11% of Lime Blue contigs and 59.31% of Poison Lake contigs), of which Gammaproteobacteria was the most abundant class for both (Fig. 1c). For Poison Lake, the order of phototrophic sulfur bacteria Chromatiales was the most abundant Gammaproteobacteria (reads: 11.38%; contigs: 8.60%, Fig. 1d). Within the order Chromatiales, Poison Lake water samples show higher relative abundances of families *Chromatiaceae* (reads: 8.45%; contigs: 8.38%) and *Ectothiorhodospiraceae* (reads: 1.98%; contigs: 1.25%) and contained a variety of PSB genera in abundances ranging from <0.25% to 3.82%, with *Thiodictyon* spp. (reads: 2.34%; contigs: 3.82%) being the most abundant (Fig. 1d). In contrast, phototrophic sulfur bacteria represented a smaller fraction of the metagenomic dataset in Lime Blue sediment, with a greater abundance of GSB from phylum Chlorobi (reads: 1.26%; contigs: 1.24%) than PSB, order Chromatiales (reads: 0.83%; contigs: 1.05%). The genera

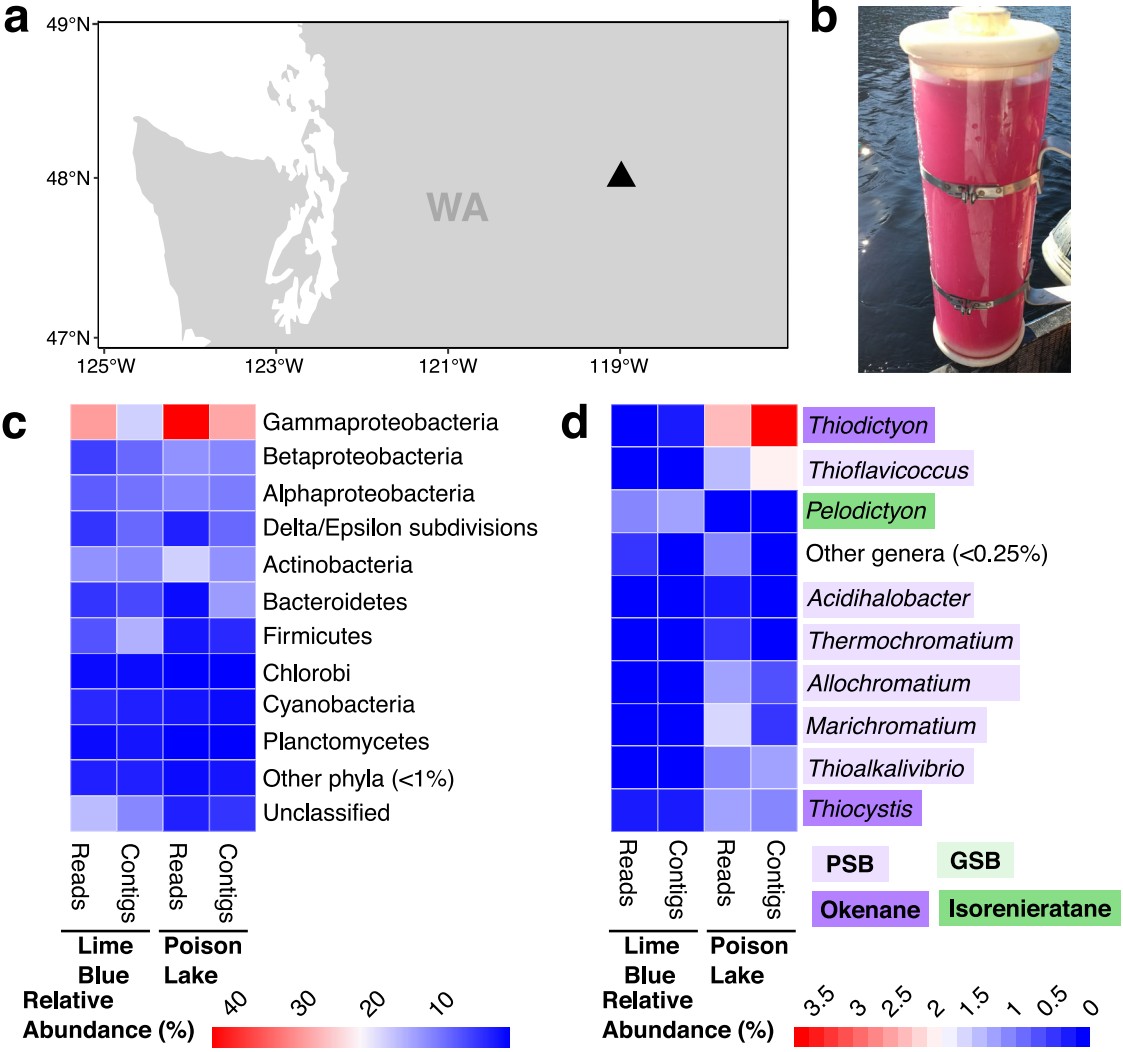

**Fig. 1 Sampling sites and bacterial community composition of the two study sites. a** Geographic location of sampling sites Poison Lake and Lime Blue in Washington (WA), U.S. **b** Poison Lake water sample displaying PSB bloom. **c** Phylum-level read and contig abundances with members of the phylum Proteobacteria split into class level, and (**d**) genus-level abundance of Chlorobi, *Chromatiaceae* and *Ectothiorhodospiraceae*. The two sampling sites are denoted as LB, for Lime Blue, and PL for Poison Lake. PSB are highlighted in purple, GSB are highlighted in green, and known producers of the pigments that are precursors of diagenetic products are denoted as indicated in the figure legends. In the heatmaps shown in (**c**) and (**d**), relative abundances increase from blue to red.

*Pelodictyion* spp. was the most abundant GSB (reads: 0.10%; contigs: 1.15%), and *Thiocystis* spp. (reads: 0.25%; contigs: 0.36%) was the most abundant PSB. Known producers of okenone, which is the biological precursor of the biomarker okenane, were present in both metagenomes, such as *Thiodictyon* sp. and *Thiocapsa* sp., an abundant PSB in Poison Lake water column metagenome[38,39]. The GSB *Pelodictyon* sp., which produces the carotenoid isorenieratene[40], was present in Lime Blue sediment.

**Bacterial metagenome-assembled genomes (MAGs)**. A total of 27 bacterial Metagenome-Assembled Genomes (MAGs) with a minimum completion of 50% and maximum contamination of 10% were binned from Lime Blue sediment and Poison Lake water assemblies (Supplementary Table 2). Most MAGs (17) were binned using the CONCOCT/MetaBAT2/MaxBin2 approach, nine bins using the NanoPhase pipeline, and one bin using LRBinner. After de-replicating the bins, 21 unique MAGs were identified, with five identical MAGs recovered from at least two of the binning strategies, as shown by their MASH average nucleotide identity (ANI) clustering (Supplementary Fig. 3). The

most abundant MAGs, quantified by mean coverage of Nanopore reads mapped to the bins using coverM, were the Poison Lake bins classified as *Thiohalocapsa* sp. (cluster10 bin, 17.05 to 22.96% relative abundance) followed by a *Desulfonatronum* sp. (PL.bin04, 6.12%). From Lime Blue, a Chloroflexota (LB_nanophase_bin80; 5.1% relative abundance) was the most abundant bin (Supplementary Table 2). 16S rRNA gene trees of the putative PSB and GSB bins and RefSeq PSB and GSB are shown in Supplementary Figs. 4 and 5 and Supplementary Data 1[63]).

**Diversity of PSB- and GSB-infecting phages**. VIBRANT identified 2742 putative phage genomes from Lime Blue contigs (100 medium-quality genomes, 24 high-quality, and two complete circular genomes) and 5806 from Poison Lake metagenomic reads, all of which were low-quality phage genome fragments. Contigs did not improve the quality of predicted phage genomes in Poison Lake, and filtered reads were utilized for further analyses. From publicly available PSB and GSB complete and draft genomes, VIBRANT identified 32 high-quality (HQ) phage genomes, 36 medium-quality (MQ), and 183 low-quality (LQ). Of

the HQ and MQ phages, 64 were from Chromatiales genomes (33 *Chromatidales* phages, and 31 *Ectothiorhodospiraceae*) (Supplementary Data 2). The majority (63) of HQ and MQ phages were classified as lysogenic, and of the eight phages classified as lytic, three were complete/circular. No Chlorobi phages were identified as lysogenic, indicating the absence of known integration enzymes in these prophages identified within their hosts' genomes. Four complete phage genomes were identified, one from the GSB *Chlorobium limicola* strain Frasassi, one from *Thiocystis violacea* strain DSM 207, and two from *Thiohalocapsa* sp. ML1 and *Halochromatium roseum* DSM 18859.

Homology matches against a database of PSB/GSB genomes predicted hosts for 5451 of the putative phage genomes (12 from Lime Blue and 5439 from Poison Lake), with the most common host in both samples being *Chromatium weissei* DSM 5161. Homology matches against MAGs resulted in 547 high-confidence predictions, with the PSB Poison Lake-bin01 (*Thiohalocapsa* sp.) and the Poison Lake-bin04 (*Desuloanatronum* sp.) as the most common hosts. High-confidence phage-host linkages based on CRISPR-spacer homology matches with 100% identity, and >20 nucleotide coverage predicted hosts for 54 phages (44 from Lime Blue and 10 from Poison Lake). The most common host for Lime Blue phages was *Ectothiorhodospira* spp., while in Poison Lake phage hosts included *Allochromatium* spp., *Chlorobium* spp. and *Thiohalocapsa* spp. Homology matches to a database of tRNA sequences yielded four host predictions, with *Thiohalocapsa* sp. ML1 being the only predicted host for three Poison Lake phages, and *Thiorhodovibrio winogradskyi* strain 6511 for one Lime Blue phage.

The Lime Blue sediment and Poison Lake water column putative phage genomes were clustered with reference viral genomes from the NCBI RefSeq based on gene-sharing distances (Fig. 2)[41]. Most Lime Blue phages and PSB and GSB phage clusters had long branch lengths, evidence of low similarity between phage genomes identified in this study and viral genomes present in databases (Supplementary Fig. 6). Several clusters were formed exclusively of Lime Blue phages. Only one cluster of Lime Blue phages was closely related to a predicted phage from PSB genomes. This may indicate that many of the phages detected in this study infect uncharacterized bacterial hosts. The database viruses most closely related to the viruses identified here infected *Chromatidales* and *Ectothiorhodospiraceae*, with the taxonomy of most hosts unresolved beyond the family level.

**Phage AMGs influencing diverse metabolic pathways.** Poison Lake and Lime Blue phages encoded 52 and 96 AMGs, respectively, representing 153 distinct KEGG pathways, including photosynthesis, sulfur metabolism and relay, pigment synthesis, Calvin Cycle, and Pentose Phosphate Pathway (PPP) (Fig. 3a). Five phages from the *Chromatidales* genomes contained AMGs involved in sulfur metabolism and relay (*cysH*, *moeB*, and *mec*). The bacterial hosts of these phages included *C. weisse* DSM 5161 (*cysH* and *mec*), *T. violacea* DSM 207 (*cysH*), *Thiospirillum jenense* DSM 216 (*moeB*), and *Allochromatium humboldtianum* DSM 21881 (*mec*). AMG-encoding phages predicted from *T. jenense* and *A. humboldtianum* were classified as temperate. A temperate phage encoding *cysH* was detected in a plasmid of *Thioalkalivibrio* sp. A phage identified in the genome of the GSB *Chlorobium limicola* strain Frasassi encoded the CP12 gene that is involved in blocking carbon fixation through the Calvin Cycle in Cyanobacteria.

AMGs involved in the light reactions of photosynthesis (*psbA* and *psbD*) were present in both Poison Lake and Lime Blue putative phages (Fig. 3b). Phage-encoded *psbA* identified in Lime

Blue clustered closely with *psbA* from *Synechococcus* phages and uncultured phages (Supplementary Fig. 7a). The predicted tridimensional structures of *psbA* encoded by Lime Blue phages and *Synechococcus* sp. were significantly similar according to FATCAT pairwise alignment (*p*-value = 0; raw FATCAT score = 448.3; 163 equivalent positions with a root square mean deviation (RMSD) of 1.10Å without twists; Supplementary Fig. 7b)[42]. A copy of the *crtF* gene, part of the okenone synthesis pathway of pigment production, was also identified in a putative Lime Blue phage (contig_6928, Fig. 3b and Supplementary Fig. 8a). This phage genome was among the top 25% most abundant in the viral community (Fig. 3c). The predicted tridimensional structures of the proteins encoded by the Lime Blue phage and the PSB *Thiocapsa roseopersicina* displayed significant structural similarity (Supplementary Fig. 8b, *p*-value = 0; raw FATCAT score = 356.21; 188 equivalent positions with an RMSD of 3.18Å without twists).

Phages also encoded AMGs involved in PPP and the Calvin Cycle. The gene G6PD/*zwf* was encoded by phages that are dominant members of the phage community in Poison Lake (blue annotation in Fig. 3d). Phylogenic analyses of the amino acid sequences of G6PD and publicly available homologous proteins from phages and bacteria showed that phage-derived Poison Lake G6PD proteins clustered with those encoded by members of *Chromatiaceae*, such as *Thiohalocapsa* spp. and *Halochromatium* spp. (Fig. 4a). A similar pattern was observed in the canonical G6PD encoded by *Synechococcus* spp. and its phages' AMGs. The predicted structures of G6PD from a Poison Lake phage and *Thiohalocapsa* sp. ML1 were compared via pairwise structural alignment (Fig. 4b). Despite the phage-encoded G6PD being shorter than *Thiohalocapsa* sp. ML1, the two structures displayed significantly similar FATCAT alignment with a *p*-value of $2.63 \times 10^{-7}$, and 249 equivalent positions with an RMSD of 3.02Å and 1 twist (via the flexible alignment procedure).

Lime Blue phages encoded several AMGs involved in sulfur metabolism (*cysE*, *nrnA*, and *pshA*) and sulfur relay (*moeB*, *thiF*, and *iscS*). While most of the AMGs were detected in phages predicted to be lytic, four Lime Blue temperate phages contained a copy of *cysH*, *moeA*, and *nrnA*. No Poison Lake phages contained sulfur metabolism or relay AMGs. The CysH protein tridimensional structure was significantly similar between Lime Blue phages and the PSB *Thiocapsa roseopersicina* (Supplementary Fig. 9a, b; *p*-value = $1.85 \times 10^{-10}$; raw FATCAT score = 356.21; 188 equivalent positions with an RMSD of 3.18Å without twists).

Among the phages with AMGs involved in pigment production, carbon and sulfur metabolisms, three Lime Blue phage-host linkages could be made with high confidence based on CRISPR-spacer homology matches, two were predicted to infect the GSB *Chlorobium chlorochromatii* CaD3 (encoding *moeB* and *iscS*), and one predicted to infect *Pararheinheimera soli* BD-d46 (encoding *nrnA*). From the lower confidence matches (100% identity, 18–20 nucleotide coverage, and <2 mismatches), we identified nine Lime Blue phage-host pairs among the phages with AMGs of interest. This included a *crtF*-containing Lime Blue phage (contig_6928) predicted to infect the PSB *Thiocystis violascens* DSM 198, a temperate phage with two copies of *cysH* (contig_11073) predicted to infect the GSB *Chlorobium phaeobacteroides* DSM 266, and a phage encoding *thiF* (contig_43205) infecting the PSB *Arsukibacterium* sp. MJ3. Protein phylogeny of the translated CrtF protein with publicly available bacterial and viral proteins demonstrate clustering of the phage-encoded protein with the host-encoded protein (Supplementary Fig. 8a).

The rank-abundance curves displaying the relative abundances of phages encoding AMGs differ substantially between the two metagenomes. Viruses encoding AMGs of interest from Poison Lake present in the top 23% ranks of viral genomes recovered

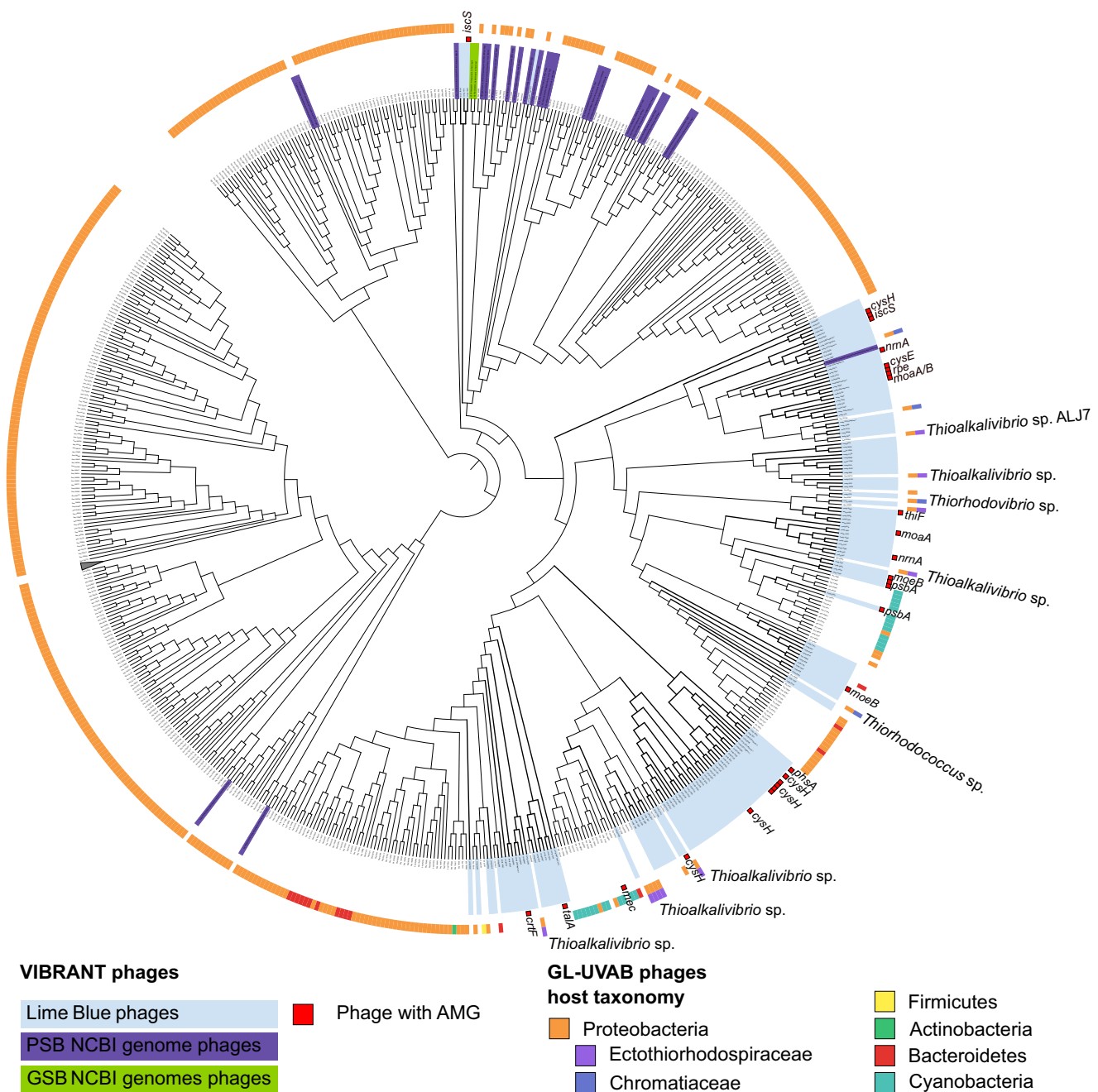

**Fig. 2 Phylogenomics of Lime Blue phages, phages identified in cultured PSB and GSB genomes from the NCBI RefSeq, and phages from the GL-UVAB database.** The VIBRANT phages from Lime Blue, many of which contained AMGs of interest (inner ring), form branches with low similarity to reference phage genomes (light blue branches). Where known, the host genus of the reference PSB-infecting phage was indicated (outer ring). The branch lengths are ignored to better display clustering topology. For a version displaying branch length, see Supplementary Fig. 6.

from the metagenomic dataset. In contrast, in Lime Blue sediment, the AMGs are present across the entire rank-abundance curve (Fig. 3c, d). The top three Lime Blue sediment phages with AMGs of interest encoded *thiF* (rank 76), *psbA* (rank 119), and *cysH* (rank 1918) (Fig. 3c). Genes involved in sulfur relay and metabolism were present in viruses across multiple ranks, between viruses at rank 76 to 1574. The majority of the AMGs involved in the other three metabolic processes were largely present in the top 401 ranks. In Poison Lake, the AMGs of interest were encoded by phages located between ranks 133 and 1875 for *psbA* and *cysH*, respectively (Fig. 3d). Overall, the AMGs of interest were encoded by viruses that constitute the top 50% of phages identified in the metagenomes.

## Discussion

Here, we report putative viral genomes recovered from Lime Blue and Poison Lake, two euxinic lakes in the Pacific Northwest. Long-read metagenomes included previously undescribed viral lineages infecting GSB and PSB, as evidenced by the long branch lengths in phylogenomic trees (Supplementary Fig. 5). Many of these phages encode AMGs with the potential to modify hosts' metabolism and ecology. Based on these results, we propose that bacteriophages have the potential to affect the metabolism and ecology of GSB and PSB by modulating (a) the synthesis of light-harvesting molecules, (b) carbon fixation, and (c) sulfur metabolism (Fig. 5).

The photosynthetic apparatus of non-oxygenic bacteria consists of light-harvesting protein-pigment complexes, which use

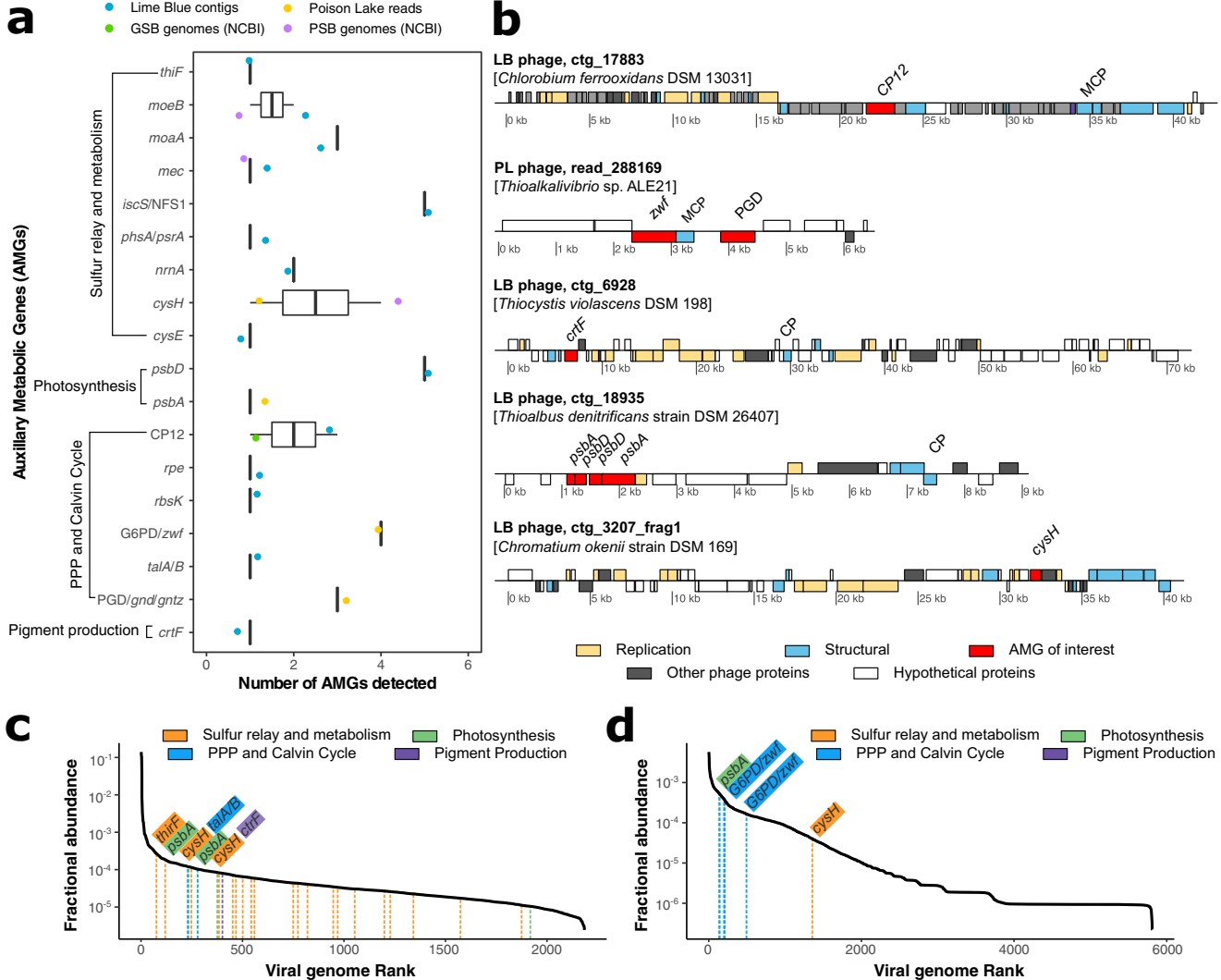

**Fig. 3 Distribution of phage auxiliary metabolic genes (AMGs). a** AMG abundances (median and standard deviation) and (**b**) genome maps of putative phages containing AMGs of interest. AMGs were identified by VIBRANT except for CP12, which was identified by BLASTp of phage ORFs to a database of available CP12 proteins from UniProt. Putative hosts identified based on CRISPR spacers are indicated for each phage. Viral rank-abundance curves for (**c**) Lime Blue and (**d**) Poison Lake. Each putative phage genome identified represents a point on the rank abundance curve and phages encoding AMGs of interest are annotated with the dotted lines. (LB Lime Blue, PL Poison Lake, MCP major capsid protein, CP putative capsid protein, PGD 6-phosphogluconate dehydrogenase [*zwf*]).

carotenoid and bacteriochlorophyll as primary donors. The diagenetic products of light-harvesting pigments (i.e., chlorobactene, isorenieratene, and okenone) preserved in sediments and in the geologic record are used as proxies of the photic zone euxinia[18,20]. However, previous studies have shown a decoupling between the abundance of GSB and PSB and the concentrations of their pigments in sediments of modern environments[21,23,24]. These observations suggest that other biological controls may be at play. Based on our metagenomes from Lime Blue and Poison Lake, we suggest that viral infections modify the production of protein-pigment complexes by bacteria, affecting their geochemical signal.

We identified a phage encoding a gene for the second-to-last step in okenone synthesis (*crtF*)[43] and predicted to infect the PSB *T. violascens* DSM 198 (Fig. 3b). We hypothesize that this phage gene may increase the production of okenone by PSB during viral infection. Additional okenone may increase rates of light reactions of photosynthesis, accelerating ATP production for viral particle assembly. This mechanism is similar to that observed in phages that increase rates of light reactions in Cyanobacteria[34].

This increase in okenone production could potentially explain the higher relative abundance of okenone in Lime Blue despite the dominance of GSB in this lake. Previous work showed that horizontal gene transfer in Lake Banyoles (Spain) results in the unexpected synthesis of photosynthetic pigments (bacteriochlorophyll e and isorenieratene) by green-pigmented GSB, *Chlorobium luteolum*, a bacterium that usually synthesizes bacteriochlorophyll c[27]. This gene transfer event offered a fitness advantage to *C. luteolum* over brown-pigmented GSB by the expansion of its photo-adaptation range to a deeper photic zone. This example of Lake Banyoles is evidence that exogenous genes acquired laterally may affect pigment production, supporting the idea that phage genes in Lime Blue may affect pigment synthesis in PSB.

We also identified putative viral genomes carrying genes (*psbA, psbD*) that encode key photosystem II proteins (D1, D2) in PSB and GSB. The discovery of these genes in the genomes of phages that infect Cyanobacteria in modern oceans suggested phage-encoded proteins have a direct role in determining the rates of light reactions of photosynthesis in the ocean and thereby, oxygen

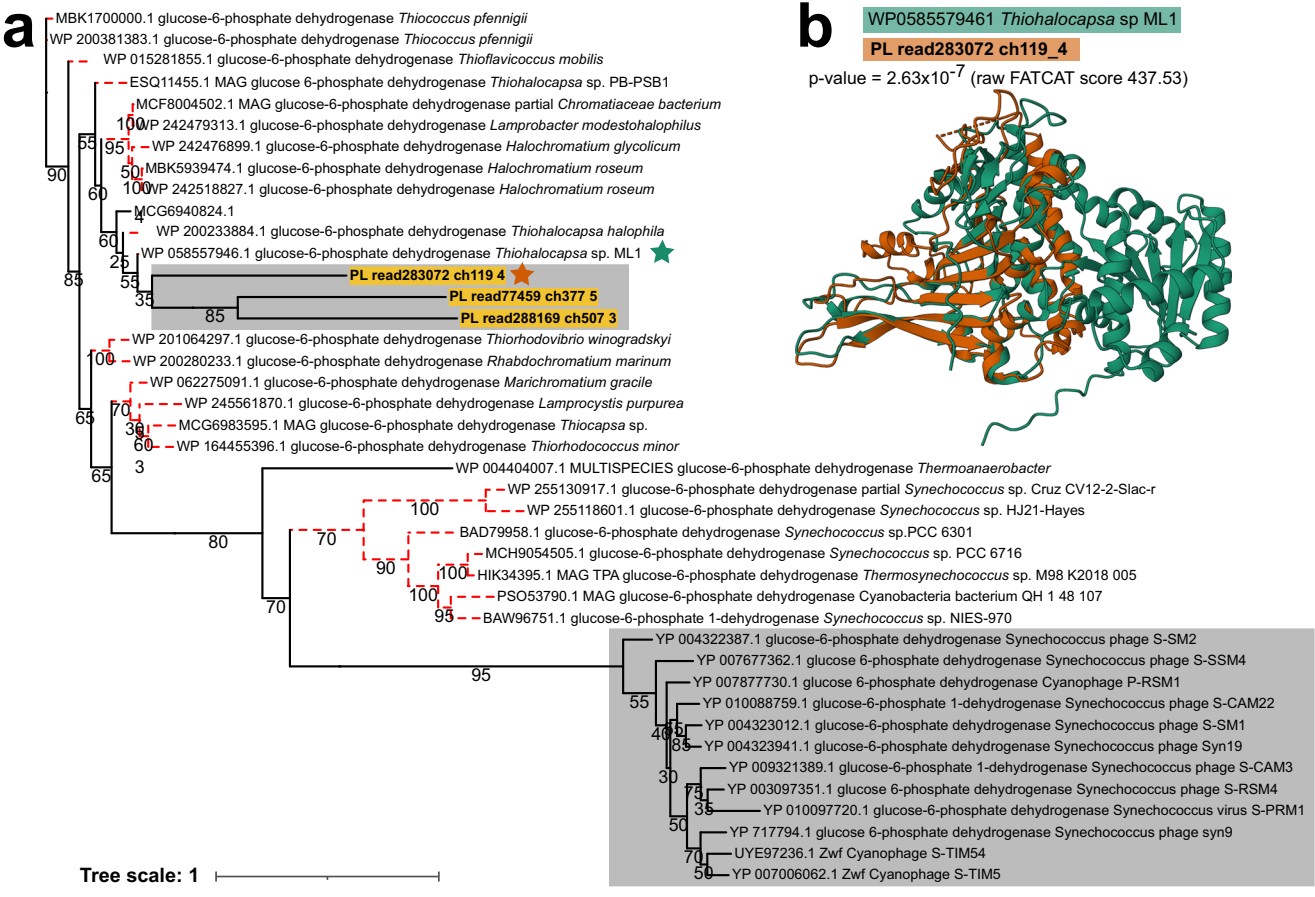

**Fig. 4 Relationship between glucose-6-phosphate dehydrogenase from phage genomes identified in this study and RefSeq non-redundant phage and bacterial proteins. a** Maximum-likelihood phylogeny of 41 glucose-6-phosphate dehydrogenase amino acid sequences (G6PD). Red dotted branches indicate bacterial proteins and grey boxes highlight phage proteins. **b** the superimposed protein structure is the result of a pairwise comparison of G6PD proteins from *Thiohalocapsa* sp. ML1 (green) and Poison Lake phage read283072 (orange). Folded proteins are denoted by a star on the phylogenetic tree with the aforementioned colors, and *p*-value of the alignment and raw FATCAT score are reported within the figure.

production[44]. In Lime Blue, by modifying light reaction rates through the expression of these genes, phage infection could indirectly affect the metabolism of pigment molecules associated with reaction centers. Simply put, viral infections could increase the production of light-harvesting molecules and accelerate rates of ATP production used in viral particle assembly. Viral-mediated changes in biomarker abundance would need to be considered when using pigment biomarkers as indicators of photic zone euxinia depth. For instance, viral infection could lead to higher okenone production and consequent okenane preservation in the sediments. This would lead to an overestimation of PSB and, therefore, an inaccurate interpretation of shallow photic zone euxinia.

The contribution of PSB and GSB to photosynthetic production in euxinic lakes is proposed to be differentiated using the carbon isotope composition of organic matter ($\delta^{13}C_{org}$). PSB and GSB fix carbon utilizing different enzymatic pathways that fractionate carbon isotopes to different extents, producing $\delta^{13}C_{org}$ values in PSB that are lower than those of GSB using the same carbon source[23,45]. However, PSB and GSB contributions to carbon fixation are not always correlated with their abundance, as demonstrated in Lake Cadagno, Switzerland[46]. We propose that PSB and GSB viral infections that modulate rates of dark reactions of photosynthesis could explain this pattern (Fig. 5). In Cyanobacteria, phage infections alter not only light reactions but also the Calvin Cycle, the Pentose Phosphate Pathway, and nucleotide biosynthesis through the expression of AMGs (e.g., *rpi*,

*talC, tkt*, and *can*)[32]. Specifically, viral infections can shut down carbon fixation while maintaining or even supplementing light reactions and the production of pentoses to support phage replication[29,34,47–54]. Cyanobacteria share with PSB and GSB the reductive pentose phosphate and reverse tricarboxylic acid cycle pathways utilized for carbon fixation, and PSB also uses the Calvin Cycle[55,56]. Viruses encoding genes that modulate carbon fixation were present among the 500 most abundant viral genomes in the Poison Lake dataset (Fig. 3d). In both lakes, we identified phages encoding AMGs capable of blocking the Calvin Cycle (CP12) and upregulating the Pentose Phosphate Pathway (*gnd, zwf, tal*) and the synthesis of reaction centers (*psb*). These genes were encoded by phages predicted to infect PSB (Figs. 2, 3). These observations suggest that carbon isotope fractionation associated with carbon fixation rates by anoxygenic phototrophs can be modified (up or down) if viral strains encoding these AMGs are actively infecting.

Phototrophic sulfur bacteria oxidize inorganic sulfur compounds under anaerobic conditions. All phototrophic Chromatiaceae, most Ectothiorhodospira, and GSB oxidize sulfide and elemental sulfur to sulfate, using them as electron donors for photosynthesis[57]. The combined effects of microbial sulfide oxidation, sulfate reduction, and disproportionation generate an apparent fractionation between isotopes of sulfate and sulfide ($\Delta^{34}S = \delta^{34}S_{sulfate} - \delta^{34}S_{sulfide}$)[57–61]. Therefore, the isotopic product-reactant discrimination in modern environments and rock records are interpreted as microbial processes that induce

## a Light-harvesting molecules

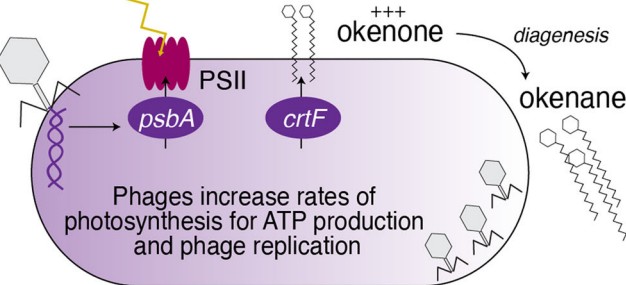

## b Carbon fixation

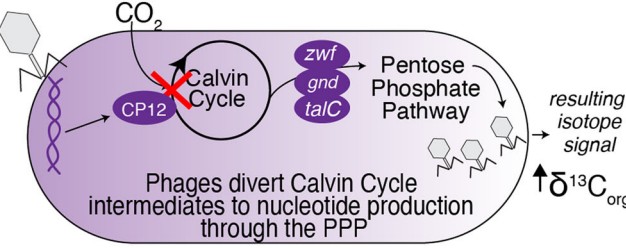

## c Sulfur metabolism

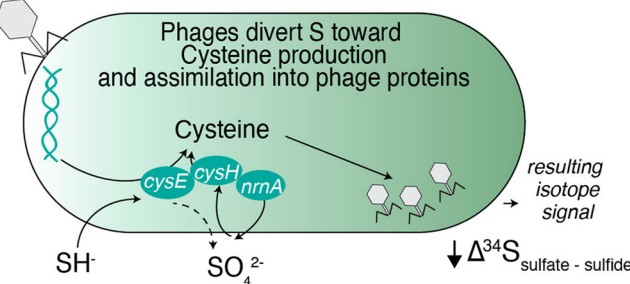

**Fig. 5 Conceptual hypotheses for viral infection influence on PSB and GSB communities.** Viral infection and gene transfer affect the biosignatures of PSB and GSB by modulating their (**a**) pigment production, (**b**) carbon fixation, and (**c**) sulfur metabolism.

sulfur isotope fractionations. The $\delta^{34}S$ fractionations associated with phototrophic sulfur oxidation are proposed to be correlated with photosynthetic activity[16] and the sulfur flow through the bacterial metabolism[59,60].

In Lime Blue and Poison Lake, we identified nine phage genomes encoding genes involved in sulfur metabolism and relay system (Fig. 3), including genes involved in sulfur assimilation as cysteine (*cysH, mec*) and genes involved in the synthesis of molybdopterin, a coenzyme participating in many pathways for sulfur and nitrogen metabolism[62]. The majority (22) of the AMGs involved in sulfur relay and metabolism are encoded by both dominant and rare viruses distributed across the rank-abundance curve (Fig. 3c, d). The gene *cysH*, involved in the oxidation of inorganic sulfur compounds, has also been observed in single-cell genomes of viruses infecting the GSB *Chlorobium clathrtiforme* from a stratified gypsum karst lake in Lithuania[63].

We hypothesize that phages divert sulfur from the bacterial energetic metabolism (photosynthesis) towards amino acid synthesis for viral particle production. Such drift in the sulfur flow has the potential to modify sulfur isotopic fractionation. The presence of the genes *cysE* (cysteine biosynthesis) and *cysH* (assimilatory sulfate reduction) in putative phage genomes predicted to infect PSB in Lime Blue supports this hypothesis (Fig. 3). *cysE* (serine O-acetyltransferase) is required in the amino

acid cysteine synthesis pathway from serine and sulfide. The expression of phage *cysE* during infection may, therefore, shunt sulfide from the oxidation pathway associated with the light reactions of photosynthesis toward the increased production of cysteine directed at viral protein synthesis (Fig. 5). This would result in a decrease in the sulfur isotope fractionation between sulfate and sulfide in infected cells. Likewise, *cysH* encodes a reductase that catalyzes the conversion of phosphoadenosine phosphosulfate (PAPS) to sulfite. This enzyme is typically repressed during photoautotrophic growth using hydrogen sulfide as an electron donor and is used to incorporate sulfate into amino acids[64]. The expression of phage-encoded *cysH* could increase the supply of sulfite consumed by Mo-containing enzymes, cascading to increased cysteine synthesis and, presumably, a decrease in the difference between sulfur isotope fractionation between sulfate and sulfide. Our observations introduce the potential for applying isotope data to infer viral effects on microbial sulfur cycling.

The current study focuses on two euxinic lakes. However, the presence of phages encoding the AMGs of interest in publicly available genomes of PSB and GSB isolated from other lakes, sediments, freshwater creeks, and coastal seawater around the world (Fig. 2 and Supplementary Data 1[63]) suggests a broad distribution and significance of these viral genes. Future work is needed to demonstrate active viral infections in the lakes studied and whether viral gene expression during infection alters host metabolic pathways as predicted here. While viral isolates encoding AMGs are not currently available, mesocosm experiments manipulating bacterial and viral densities and quantifying rates of carbon fixation, pigment production, sulfur oxidation coupled with transcriptomics will shed light on the active viruses and their AMGs. Incorporating both size-fractionated cellular metagenomes, viromes, and proximity ligation sequencing approaches will be essential to identifying active and dormant prophages within the viral community[65]. Ultimately, the isolation of AMG-encoding viruses infecting PSB and GSB will enable genetic manipulation for functional validation.

## Conclusion

Here, we describe PSB- and GSB-infecting putative viral genomes from modern euxinic lakes, microbial ecosystems that shed light on the ecology of primary producers in Earth's deep time. These phages encode metabolic genes with the potential to regulate pigment production, photosynthesis, carbon fixation, and sulfur metabolism, suggesting that these viruses can affect host physiology and ecology. Our observations suggest that viral infections could impact biosignatures of phototrophic sulfur bacteria in the sedimentary record.

## Methods

**Study sites.** The research was conducted in two shallow (<16 m), sulfidic lakes: Lime Blue and Poison Lake in eastern Washington, U.S. (48°N, 119°W, Fig. 1a). The study sites are closed-basin lakes that only lose water by evaporation and seepage and receive water from direct precipitation, runoff, and catchment groundwater[66]. Undeveloped catchments, strong salinity gradients, and closed-basin configurations promote the prolonged periods of meromixis and benthic euxinia required by PSB and GSB, making these lakes ideal study sites.

**Sampling.** Poison Lake and Lime Blue water chemistry were characterized in the field using an HYDROLAB Multiparameter Sonde (OTT, Germany) and sulfide concentrations were measured concurrently using the Cline method[67], and a DR 2800 field spectrophotometer (Hach, CO). Their vertical oxygen and sulfide profiles are shown in Supplementary Fig. 1. Poison Lake water (2 L) from the sulfidic zone (6.5 m depth) was collected from a boat using a peristaltic pump (Fig. 1b). Subsamples (50 ml) for microbiology analyses were immediately frozen until further laboratory processing. In the laboratory, samples were defrosted and incubated overnight at 4 °C with Polyethylene Glycol 8000 10%. The samples were centrifuged at 5000 g for 2 h at 4 °C and the pellet containing both viruses and bacteria was extracted for DNA with a DNeasy PowerSoil kit (Qiagen, Germany)[68]. The sediment from Lime Blue was sampled using a freeze core[38]. The sediments were

sectioned within a sterile flow hood to prevent organic contamination. Sediment from the top 2 cm (1 g) was extracted using the DNeasy PowerSoil kit (Qiagen, Germany) without size fractionation and following the manufacturer's instructions.

**Long-read metagenomic sequencing.** Poison Lake and Lime Blue metagenomic libraries were prepared using the ONT Ligation Sequencing Kit (SKQ-LSK110, Oxford Nanopore Technologies, UK) following the manufacturer's instructions. In short, DNA quality was assessed by fluorometry using Qubit 2.0 (Invitrogen, USA) using the dsDNA High-Sensitivity Assay. Metagenomic dsDNA (>1 µg) was End-prepped and repaired to ligate a poly-A tail using the NEBNext Companion Module for Oxford Nanopore Technologies Ligation Sequencing (cat # E7180S) before sequencing adaptors were ligated onto the ends. Between each step, DNA was cleaned using 60 µl Agencourt AMPure XP beads (Beckman, USA), washing the beads with 70% molecular grade Ethyl alcohol (Sigma-Aldrich, USA) before resuspending in 61 µl Nuclease-free water (Fisher, USA). Sequencing libraries were sequenced using a FLO-MINSP6 flow cell (R.9 chemistry, Oxford Nanopore Technologies, UK), and the sequencing protocol was run for 48 hrs.

**Generation and quality control of MAGs.** Sequencing adaptors were trimmed using Porechop v0.2.4[39,40] and trimmed reads were assembled with Flye v2.9[69,70] using the --meta parameter. In parallel, low quality and short reads were removed by NanoFilt v2.6.0[71] to a minimum Q-value of 9 and length of 1 Kb. Metagenome-assembled genomes (MAGs) of bacteria were generated through three strategies. In the first, hight quality-controlled reads were mapped to metaFlye contigs with Minimap2. The SAM files were compressed, sorted, and indexed with samtools v1.9[72]. Metagenomic bins were generated using a combination of three binning programs: MetaBAT2 v2.12.1[73], MaxBin2 v2.2.6[74] as previously described[75], and CONCOCT v1.0[76]. The resulting bins were refined using MetaWRAP v1.3 bin_refinement module[77] and refined bins were assessed for contamination and completion with CheckM v1.2.0[78]. In the second approach, the binning program LRBinner v.2.1[79], which is specialized in long reads, was utilized to bin metagenomic contigs. The third approach applied the long-read binning pipeline Nano-Phase v.0.2, which utilizes MetaBAT2 and MaxBin2, and has been validated on the ZymoBIOMICS gut microbiome standard[80]. All bins with ≥50% completion and ≤10% contamination were kept for further analyses[81]. MAG depth of coverage (mean) was quantified by mapping quality-controlled reads to the metagenomic bins and taking the mean percentage of reads mapped with the tool coverM v0.6.1[82]. Finally, duplicate MAGs from different binning approaches were identified using dRep v.3.0.0[83].

**Taxonomic profiles of lake bacteria.** ONT reads and contigs were taxonomically classified by Kraken v2.0 and abundances were estimated by Bracken (Bayesian Re-estimation of Abundance after Classification with KrakEN) v2.738 using the RefSeq database (accessed March 2022)[84,85]. The taxonomy of MAGs was determined using GTDB-Tk v.2.1.1 (accessed November 2022) using the classify workflow (classify_wf)[86,87].

**Identification of viruses in metagenomes and PSB and GSB genomes.** Both the metaFlye contigs and high-quality ONT reads were utilized for the detection of phages by VIBRANT v1.2.1, a bioinformatics pipeline that uses Hidden Markov Model (HMM) searches to identify clusters of viral genes in unknown sequences, allowing the sorting of high-confidence viral genomes and genome fragments within complex samples[88]. To obtain the abundance and coverage of putative viral genomes in the environment, trimmed reads were mapped with Minimap2 v2.24[89] to the viral contig database at high stringency (>95% identity)[90].

Publicly available bacterial genomes deposited as 'complete genome', 'scaffold', or 'contig' belonging to the two PSB families *Chromatiaceae* (98 genomes) and *Ectothiorhodospiraceae* (115 genomes), and the GSB phyla Chlorobiota (33 genomes) were retrieved from NCBI in 2022 (accession numbers available in Supplementary Data 1[63]). Putative prophages were identified in these genomes using VIBRANT v1.2.1. A summary of the data generated and utilized for the purpose of this study can be found in Supplementary Table 1. Viral genomes identified within bacterial genomes from the RefSeq were identified as temperate.

Phylogenomic analysis of phages identified in this study was performed against the GL-UVAB (Gene Lineage of Uncultured Viruses of Archaea and Bacteria) database, using the script (GLUVAB.pl) described within the publication[41]. A summary of the entire workflow is shown in Supplementary Fig. 2, and a summary of the phage genomes identified is provided in Supplementary Data 2[93].

**Phage host prediction.** Viral hosts were identified using a combination of gene homologies, the presence of tRNAs, and CRISPR (clustered regularly interspaced short palindromic repeats) spacers[41,91]. (I) Sequence homology matches were made from the phages identified from Lime Blue and Poison Lake to databases generated from PSB and GSB genomes retrieved from NCBI and MAGs generated in this study using BLASTn[92]. Only hits >80% sequence identity across a minimum alignment of 1000 nucleotides were considered as putative hosts for NCBI and RefSeq genomes, and 95% sequence identity against MAGs, as previously described[41]. (II) A database was created with the CRISPR spacers from PSB, GSB genomes and MAG using minCED v0.4.3 (Mining CRISPRs in Environmental

Datasets), which uses CRISPR Recognition Tools (CRT) v1.2[93,94], and sequence homology matches were made against the phages using BLASTn with the parameter -task "blastn-short", hits were only considered with a maximum of 2 mismatches or gaps, 100% coverage to spacer, and minimum length of 20 nucleotides, as described in previous work[41,95]. (III) Phage tRNAs were detected using tRNAScan-SE v2.0[96], and matched against PSB/GSB/MAG genomes using BLASTn at ≥90% sequence identity and ≥ 90% coverage, as described in previous work[41].

*Analysis of auxiliary metabolic genes.* VIBRANT identifies viral auxiliary metabolic genes (AMGs) and viral genomes' potential for lysogeny (presence of transposases and integrases) through HMM comparisons with three databases: Kyoto Encyclopedia of Genes and Genomes (KEGG) KoFam (March 2019 release)[97–99], Pfam (v.32)[100,101], and Virus Orthologous Groups (VOG) (release 94). VIBRANT utilizes a manually-curated collection of viral AMGs from KEGG annotations falling under the metabolic pathways and sulfur relay system categories. The AMG outputs from VIBRANT were manually curated for carbon, sulfur, and pigment-related AMGs. Viral genomes containing AMGs of interest were visualized using the R package genoPlotR v0.8.11[102]. For ten phages containing AMGs of interest, the Max Planck Institute (MPI) HHpred server was utilized to manually improve genome annotations (E-value < 0.01 and Probability > 80%)[103], in addition to the Phage Artificial Neural Networks (PhANNs) to confirm phage structural proteins (Confidence > 80%)[104].

Protein phylogeny was performed on four viral AMGs of interest (*psbA*, G6PD, *crtF* and *cysH*) and homologous viral and bacterial proteins from the RefSeq. Proteins were first de-replicated at 99% identity using CD-HIT v.4.8.1[105], before alignment with MAFFT v.7.508[106,107]. Maximum-Likelihood phylogenetic trees were constructed with RAxML-HPC v.8.2.12[108], with the PROTGAMMAAUTO parameter allowing RaxML to calculate the best substitution model for each dataset and 200 bootstrap repetitions. Resulting trees and bootstrapping values were visualized with the Interactive Tree of Life v6 (iTOL)[109,110]. Predicted viral AMGs and their closest relative according to the protein phylogenies were folded using AlphaFold through ColabFold[111,112]. Protein structures were compared using FATCAT2 (Flexible structure AlignmenT by Chaining Aligned fragment pairs allowing Twists) pairwise alignment to acquire similarity values[42]. Aligned proteins structured using FATCAT2 were considered to have structural relationship with an alignment *p*-value < 0.1, with lower values indicating higher similarity.

**Reporting summary.** Further information on research design is available in the Nature Portfolio Reporting Summary linked to this article.

## Data availability

The Nanopore metagenomic sequencing data generated here are available in the Sequence Reads Archives (SRA) repository under the BioProject PRJNA842402: Lime Blue sediment (SRS13178833) and Poison Lake water (SRS13178834). Datasets are provided as csv files through Figshare (https://figshare.com/projects/Viruses_of_green_and_purple_sulfur_bacteria/162820), including access codes for purple and green sulfur bacteria genomes retrieved from the National Center for Biotechnology Information (NCBI) RefSeq (Supplementary Data 1)[113], a complete list of predicted phage-hosts pairs, phage genome quality, and phage AMGs (Supplementary Data 2)[114], and separate csv files for data plotted in Figs. 1 and 3.

## Code availability

The codes used for bioinformatic analyses[115] are available through Figshare (https://figshare.com/projects/Viruses_of_green_and_purple_sulfur_bacteria/162820).

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

## Acknowledgements

We would like to thank James Harris, James Fulton, and Byron Steinman for contributing to the water column and freeze core collection from Lime Blue. This research was supported by the NASA Exobiology Program (80NSSC23K0676 to C.B.S., A.B.S., W.P.G., and J.P.W.). Samples were collected through funding from a Purdue Research Foundation Research Grant to W.P.G. and National Science Foundation grants to J.P.W. (EAR-1424170) and W.P.G. (EAR-1424228). Computational analyses were funded by the University of Miami Institute for Data Science and Computing – Expanding the Use of Collaborative Data Science to C.B.S.

## Author contributions

P.J.H.B.: genomic analyses and data visualization; A.B.S.: study design, sampling, sample processing; S.L.G.: metagenomic sequencing; M.D.O'B.: sample collection and processing; J.P.W.: sample collection, processing, and funding; W.P.G.: sample collection, processing, and funding; C.B.S.: study design, analyses, funding, and writing. A.B.S., P.J.H.B., and C.B.S. wrote the first version of the manuscript, and all authors contributed to revisions.

## Competing interests

The authors declare no competing interests.

**Additional information**

