## [Peer Review File · Communications Earth & Environment]

18th Oct 22

Dear Dr Silveira,

Please accept my apologies for the delay in obtaining reports on your manuscript titled "Genomics of viruses infecting green and purple sulfur bacteria in two euxinic lakes". Your manuscript has now been seen by 4 reviewers, whose comments are appended below. You will see that they find your work of some potential interest. However, they have raised quite substantial concerns that must be addressed. In light of these comments, we cannot accept the manuscript for publication, but would be interested in considering a revised version that fully addresses these concerns.

We hope you will find the reviewers' comments useful as you decide how to proceed. Should additional work allow you to:

- address these criticisms (that is, either to incorporate the suggestions or provide a compelling argument why the point made by the reviewer is not valid, or relevant to the editorial threshold as outlined below)

AND

- meet our editorial thresholds as outlined below,

then we would be happy to look at a substantially revised manuscript.

In the following, we list our main editorial thresholds:

- Provide compelling new insights into the effects of viral infections on host green and purple sulfur bacteria in anoxic and sulfidic lakes;
- Provide comprehensive method details and demonstrate that your approach is robust and valid;
- Provide compelling support for your claims regarding the environmental importance of viral infection in communities of sulfur bacteria or, alternatively, tone down your claims and describe the future work required to support the hypotheses drawn from your findings.

In addition, please supply point-by-point responses to all the reviewers' comments.

If you choose to take up this option, please either highlight all changes in the manuscript text file, or provide a list of the changes to the manuscript with your responses to the reviewers.

If the revision process takes significantly longer than three months, we will be happy to reconsider your paper at a later date, as long as nothing similar has been accepted for publication at Communications Earth & Environment or published elsewhere in the

meantime.

We understand that due to the current global situation, the time required for revision may be longer than usual. We would appreciate it if you could keep us informed about an estimated timescale for resubmission, to facilitate our planning. Of course, if you are unable to estimate, we are happy to accommodate necessary extensions nevertheless.

Please use the following link to submit your revised manuscript, point-by-point response to the reviewers' comments with a list of your changes to the manuscript text (which should be in a separate document to any cover letter) and any completed checklist:

[link redacted]

Please do not hesitate to contact me if you have any questions or would like to discuss the required revisions further. Thank you for the opportunity to review your work.

Best regards,

Clare

Clare Davis, PhD
Senior Editor
Communications Earth & Environment

www.nature.com/commsenv/
@CommsEarth

EDITORIAL POLICIES AND FORMAT

If you decide to resubmit your paper, please ensure that your manuscript complies with our editorial policies and complete and upload the checklist below as a Related Manuscript file type with the revised article:

Editorial Policy Policy requirements (Download the link to your computer as a PDF.)

For your information, you can find some guidance regarding format requirements summarized on the following checklist:(<https://www.nature.com/documents/commsj-phys->

style-formatting-checklist-article.pdf) and formatting guide (<https://www.nature.com/documents/commsj-phys-style-formatting-guide-accept.pdf>).

REVIEWER COMMENTS:

Reviewer #1 (Remarks to the Author):

The authors report on the diversity and distribution of viral sequences in two euxinic lakes in the US by means of long-read sequencing of viral fraction (though this is my guess, as information about sample size fractionation isn't provided) in samples from lake water and sediments. In addition, authors analyzed available sequenced genomes of green and purple sulfur bacteria to identify the presence of sequences of viral origin. The manuscript is based on the descriptive analysis of 2 samples, which is the biggest limitation of this study. Overall, this is a clear, concise, and well-written manuscript. It describes intriguing findings and, surely, can significantly contribute to the improving of our understanding about the diversity of viruses infecting sulfur metabolizing bacteria. The methods used are appropriate and provide sufficient information to be repeatable. Results and discussion are centered round taxonomic classification of identified viral sequences, which is the major novelty of this study, and description of the potential role of viral AMGs, which is only speculative as no experimental evidence could be provided. Authors found several AMGs in viral contigs, potentially involved in carbon and sulfur metabolism of their hosts and based on these findings conclude that viruses might affect host physiology and biogeochemical cycling of sulfur. This conclusion is in agreement with other recent findings and suggestions provided by Berg et al. 2021 ISME 15: 1569-84 and Šulčius et al. 2021 Genes 12: 886.

Reviewer #2 (Remarks to the Author):

Summary

The manuscript uses samples collected from two euxinic lakes combined with metagenomics to show how viral infections affect green and purple sulfur bacteria (GSB and PSB, respectively) in these proposed early Earth analog environments. The manuscript states that the results show that viral infection of GSB and PSB interferes with photosynthesis, pigment production, and carbon and sulfur metabolism. According to the manuscript, these results suggest that viral infection of GSB and PSB could impact the interpretation of their pigment biomarkers in the sedimentary record. In total, this was an interesting study, but major revisions are required prior to publication. These revisions relate to the motivation for the study and how the results impact the interpretation of GSB and PSB biomarkers in the rock record.

Major comments

1) The link between phototrophic sulfur bacteria and the GOE at the beginning of the introduction and conclusion feels forced and misaligned as a motivation for this study. For example, the oceans were likely ferruginous rather than euxinic prior to the GOE, the latter of which is more relevant for this study of euxinic lakes and GSB and PSB. The Johnston et al

reference (ref #3) cited in the first sentence is specifically about anoxygenic phototrophic bacteria dominating during the Proterozoic (post GOE) rather than Archean (pre-GOE), which conflicts with the first sentence of the introduction where it is referenced. Furthermore, the idea that GSB and PSB were significant primary producers prior to the GOE is not definitive as there is no direct evidence for it, and recent work suggests that Chlorobi may have postdated the GOE (e.g., Ward and Shih, 2022). The introduction and conclusion should be rewritten with a better justified motivation.

2) The sampling subsection of the Methods section is lacking information about the study locations. For example, where are the lakes located, why were these two lakes chosen, in what ways are these lakes similar or different (e.g., water depth, size), etc? The discussion section is the first place where the lake locations are stated, whereas this information should occur in the sampling subsection. At minimum, the main manuscript text needs to contain some brief study site details, which is currently absent.

3) The discussion section contains multiple passages that are vague and need specifics and clarification. For example, lines 324-327 present a problem where the GSB and PSB biomarker records for photic zone euxinia “are not fully explained by physical and chemical bottom-up controls”. It is not clear what this passage is specifically referring to. Next, the text states that the metagenomic results present “an alternative explanation for this observation,” but it is not clear what the “observation” is that the new results are potentially explaining. In general, the text does not state the specific problem with the GSB and PSB biomarker record that the results address or how specifically the results impact how these biomarkers are interpreted in the geologic record. Rather, the manuscript only states that “this work will have important implications” for the GSB and PSB biomarker record (e.g., lines 330-333; 403-405). What are those implications specifically? How should readers incorporate these results into using and interpreting GSB and PSB biomarkers in deep time?

Minor comments

Line 38: I suggest using another word instead of culminating. Could simply provide approximate age of GOE.

Line 39: Ocean stratification is a result of density (temperature and salinity), so this is not the correct term here.

Line 50: Another word is needed to replace intercepts since this implies that the sulfide “stops” the sunlit layer.

Line 54: Suggest “GSB and PSB pigments and their diagenetic products are biomarkers for...”

Line 55: It is not correct to state that GSB and PSB biomarkers are proxies for basin depth. Rather, they are used as an indicator for depth of intersection of photic zone and reduced sulfur, which is different from total depth of a basin or water column.

Figure 1: It would be useful to note in the figure and description which of the GSB and PSB produce the biomarkers mentioned in the text since not all GSB and PSB make the biomarker pigments.

Line 315: Suggest deleting “proxy” so the text only reads lakes, which is clearer.

Figure 4. It would be useful to illustrate in the figure (left cartoon) how viral infection affects okenone production.

Lines 333: “This hypothesis is consistent with ...” It does not seem that the hypothesis that this manuscript is proposing in lines 330-333 are either consistent or inconsistent with the

previous work described in lines 333-339, rather more accurately these results are “in addition to” the previous work showing how pigment abundance may not accurately reflect the abundance of a specific taxon. Viral infection and HGT scramble this first order relationship.

Line 352: single GSB strain rather than single GSB?

Line 381, 288: suggest using different word than deviate (e.g., divert).

Line 386: S isotopes of sulfate and sulfide or similar rather than “sulfate and sulfide isotopes”
Suggest removing the PL and LB abbreviations since they are not commonly used and do not add clarity to the text.

Typos: line 39, 395

Reviewer #3 (Remarks to the Author):

Reviewer: Dr Joanna Warwick-Dugdale

Manuscript entitled:

"Genomics of viruses infecting green and purple sulfur bacteria in two euxinic lakes"

By Dr Silveira and colleagues, submitted for publication in Communications Earth & Environment.

Summary

Overall, this study is an interesting investigation of important and novel putative bacteriophages infecting green sulfur bacteria (GSB) and purple sulfur bacteria (PSB) mined from: 1) Cellular, metagenomic long-read sequencing data obtained from the sediment and water column of two sulfidic and anoxic (euxinic) lakes; 2) publicly available GSB and PSB genomes. Having generated metagenome assembled bacterial genomes (MAGs) from their long read data, the authors examine the composition of the bacterial communities derived from their euxinic lake samples, and the diversity of mined phage genomes. A combination of approaches utilising both the long-read data generated by the authors and publicly available GSB and PSB genomes are applied for prediction of phage hosts. Lastly, putative GSB- and PSB-infecting phage auxiliary metabolic genes (AMGs) are identified, and the authors hypothesise possible effects on host metabolism. In particular, the influence of viral AMGs is highlighted as having the potential to impact the biological signatures of GSB and PSB, and thus the interpretation in of their photosynthetic activity in the geologic record.

Major claims; comments

The major claims of the paper appear to consist of the following:

- 1) The putative viral genomes mined from the cellular metagenomic long-read data infect PSB and GSB hosts (where predicted).
- 2) Such viruses include AMGs that can redirect host metabolism and pigment production.
- 3) The presence of such AMGs may influence interpretation of the geologic record when it is based on GSB and PSB biological signatures.

For the most part this paper is clearly written, however the informatics applied require some further explanation/consideration. The authors make considerable efforts to associate hosts with the novel viruses they identify from euxinic lake sediment and water. Viral AMGs of interest that may redirect host metabolism in PSB and GSB are identified, however the level of curation required to ensure that they are encoded by viruses is not evident. There is concern regarding the generation of MAGs from low quality long reads, where the tools employed have been designed and tested using short reads (i.e. reads of similar length and similarly high accuracy), and without citations or prior tests to show that the approach is valid. I also have criticisms the assessment of the possible significance of viral AMGs to host community function, and therefore interpretations of the geologic record, without: 1) knowledge of the abundance of such viruses in the environment; 2) consideration of the evidence (or lack thereof) for the functionality of the virus-encoded gene. Greater detail in the methods employed, and in some cases the logic behind choosing them (listed below), are needed to improve the reproducibility of the work. Some additional primary literature references for the methods employed (both 'wet-lab', and bioinformatic), and the effect of viral AMGs on host function, are currently missing.

Comments are listed by line and section, with the lines/sections that highlight issues of major concern in bold type (see attached Word Doc version of review for bold type).

Abstract and Introduction

Line 1: "euxinic". I would avoid use of this term in the title, as it is little used outside of the field (I would say "two sulfidic and anoxic lakes").

Line 35: "used as proxies to interpret". This sentence could be clearer; maybe "which are used as proxies for the interpretation of biogeochemical processes in early Earth Oceans".

Line 45: "pre-GOE". Acronym not specified in previous text (should be specified in line 38).

Line 46: "biotic factor controls". Viral infection is only one of the biotic factors that may contribute to the function of hosts (e.g. multicellular organisms are also biotic factors that affect host ecology).

Line 52: "as" should be "of".

Line 54: "are potential". It would be clearer to say that biomarkers etc have been used as proxies for basin depth and redox state.

Line 64 and Line 66: Better to include some primary literature here, as processes vital to arguments made later in the manuscript on the effect of viruses on host function are being introduced, e.g. (Forterre, 2012; Lindell et al., 2004, 2007; Suttle, 2007)

Line 69: "and sulfide oxidising phototrophs and". Reference for this? Also, I would exchange "and" for "thus/therefore".

Line 71: "concurrently". A better word might be simultaneously.

Line 72: "have high signatures": reads better as "High rates of horizontal gene transfer are suggested in by GSB genomes signatures".

Line 74 and 75: "Likewise [...] pcyA)": As later discussion includes the effect of phages on host ecology, it would be worth stating that the products of genes concerned with photosynthesis have been shown to actively alter rates of host photosynthesis in oxygenic hosts, and include the relevant references, e.g. (Fridman et al., 2017).

Methods

Line 87: Include depth of sulfidic zone and sample collection.

Lines 88-92: "Subsamples [...] Powersoil kit": References where this method of DNA extraction has been successfully used in previous studies (ideally on water-column community) bacterial DNA are missing.

Line 92: "freeze core": A brief description of this process, and how it has been modified from the reference provided, would improve understanding and reproducibility.

Line 93: "flow bench": Flow hood?

Line 94: "An archive section...": Not needed

Line 97-100: This is a very long sentence, and I found the meaning difficult to follow. If the 16S data is relevant, then some information on the methods employed to produce it, a representation of the results and analysis, and clearer statement of how they are important are needed here.

Line 104: "Metagenomic libraries were": Worth naming the samples the libraries were generated from here.

Line 106: "1 mg of dsDNA": Even for long-read sequencing, this seems like a lot of DNA to go into library preparation. Is this in total, or per sample? Could this amount of DNA have been recovered from 50 ml of lake water? Oxford Nanopore protocols call for assay of DNA quality prior to library preparation, as this is needed to calculate/approximate the molarity of the DNA required. Were any such assays obtained (for example, from an Agilent 'TapeStation')? This information will be needed for reproducibility.

Line 109: "using 1.8X Agencourt AMPure XP beads": This is a higher volume of beads than specified in the manufacturer's instructions (i.e., Oxford Nanopore protocol associated with the library preparation kit listed here). It should be clear where the manufacturer's protocol has been modified, and it would be even better to include the reasons for the changes. I would be interested to know why this change was made. It seems that the majority of the long-read sequences obtained by the authors were low quality and <1000 bp long (line 183). Lower bead-volume to sample-volume ratios are used to remove smaller fragments of DNA, and I wonder sticking to the original volumes specified in the protocol may help to improve read-length and quality.

Line 111: "FLOW-MINSP6". Better to quote the 'R' number as this indicates the type of ONT chemistry employed (e.g. R9.41; R10.4); Also "loaded onto and": not needed.

Line 118: "and quality": Missing word 'high' ("and high quality").

Line 130: State that VIBRANT was also used for identification of AMGs (shown in supplementary figure 1, but this step is important enough to the overall claims of paper to include here), and any references where this software has been used for the same purpose.

Line 132: State number of phages containing AMGs of interest that went on for further analysis here.

Line 135: "To analyse abundance and coverage": where is abundance data for the phages (of interest) in the environment shown/discussed?

Comment on section "Generation and quality control of MAGs" (Lines 140-150): The tools employed by the authors (e.g. CONCOCT; metaBAT2) were designed and tested with mapping of short reads for binning (i.e. the reads being mapped are all similar length and similarly high accuracy). It appears that the assumption has been made that mappings made with long reads will work equally well, despite the varied read-length and lower accuracy of long reads. This assumption requires validation using a of mock dataset to show that the approach is not creating chimeric bins, or citation of prior work where such work has been

undertaken. Additionally, based on supplementary figure 1, it's not clear why the low quality, short reads were mapped to the to the assembly for binning, which could potentially cause incorrect mapping. Why not just use the reads that are 1kb or greater in length and with a minimum Q-value of 9? Was this due to the scarcity of such reads in this dataset?

Lines 128-131: Of utmost importance for identifying viral AMG, is that those which appear on the end of a contig should not be assumed as viral. The DRAM-v paper (Shaffer et al., 2020) explains why this is the case, and the cut-offs required to ensure that an AMG really is virally encoded. If the authors have not used DRAM-V, then similar steps should have been used to curate their AMGs, neither of which currently appears in the text of the manuscript or in supplementary figure 1.

Lines: 166 176 (Section on phage host prediction: For parts I, II and II): Missing either references that have used the parameters stated, or some details on logic of choosing such.

Results

General comment after lines 181-182: "Line Blue (LB) sediment [...] "Poison Lake (PL) water." Because sediment and water column environments and the microbial communities associated with them are so distinct, I think it would improve the readability and understanding of this manuscript if the samples, MAGs and putative viruses, were always referred to with the words 'sediment' or 'water', throughout.

Line 183: Very few reads of good quality and ≥ 1 kbp length. See comment on line 109.

Line 184: Interesting that so many more bacterial contigs were generated in LB sediment assemblies than PL water, despite that fact that fewer reads were generated from the sediment samples. Could this be due to the diversity of sediment bacteria compared to the water column? It would be good to know how many OTUs were generated from each sample type so that this idea could be considered.

Line 186: Figure 1 should be mentioned here as what it shows is being compared to the Supplementary Figures.

Line 219 and Table 1. "denoted in bold". Currently no values are shown in bold type.

General comment re. section: "Diversity of PSB-infecting phages": I think an additional supplementary table that summarises the following for each source of putative phage genomes (i.e. 'LB sediment'; 'PL water'; 'Bacterial MAGs from this study'; 'Publicly available PSB and GSB genomes') would be useful in clarifying the data: The total number of long reads generated and the number that were high quality (where appropriate); the number of high/med/low quality phages predicted; total number of hosts predicted and percentage that were GSB-PSB (via each method applied); the number of AMGs predicted. It could help convey an overall sense what the dataset comprises of.

Line 235: "metagenomic sequences,". Add "generated here," before for clarity.

Line 288: "AMG Abundances". The number of AMGs detected is just that – the number. In light of the authors' arguments on phage-AMG impact to interpretation of the geologic record, vital additional information is required on the actual abundance of those AMGs in the environment. How abundant are these AMGs/the viruses that encode these AMGs? If they are not encoded by the most abundant viruses, are they likely to create a signal strong enough to impact interpretation of the geologic record?

Line 300: "putative lytic phage": Add which sample this phage came from.

Discussion

Line 311: “viral genomes recovered from LB and PL metagenomes”. At the beginning of this section I think it would be more informative to state that these are putative viral genomes, and also that they come from cellular metagenomic samples rather than viral metagenomes (as viral metagenomic data is no longer rare).

Line 315: “proxy lakes”: The word proxy doesn’t make sense here – maybe ‘model lakes’?

Line 317: “(l) light harvesting molecule production”. Awkward phrasing; maybe “production of light harvesting molecules”

Line 321: “Viral predation”: “Viral Infection” would be better as it includes the influence of lysogens.

Line 326: “record as proxies”. Add the word ‘serve’ before ‘as proxies’, for clarity.

Line 327: “by physical and chemical bottom-up controls”. Include the appropriate references again here.

Line 328: “presented an alternative explanation”. Would ‘suggest’ be a better word than ‘presented’? Also, the word ‘alternative’ is not appropriate/needed.

Lines 328-332: I have two thoughts regarding the argument presented here.

Firstly, some acknowledgement of the caution needed in the assumption of viral AMGs having the same function as host genes is required here, as cases where this is not the case are in the literature. For example, based on host function, the phage encoded nucleotide pyrophosphohydrolase MazG was previously hypothesised to play a role in phage mediated regulation of the stringent response (i.e. host reaction to nutrient deprivation) (Bryan et al., 2008). However, examination of enzymatic activity indicated that instead, cyanophage MazG is likely to enable recycling of host DNA via the hydrolysis of deoxyribonucleotides (Rihtman et al., 2019). This is not to say that viral AMG function cannot be predicted from host gene function, but it should not be assumed. The argument for caution is especially pertinent when a function is extrapolated from a single gene in a synthetic pathway of many steps (as argued by the authors of the manuscript).

Secondly, the abundance of a viral AMG (or phage that encodes an AMG) is needed for evaluation of its possible importance to the ecosystem function. Here, knowing the abundance of the viral AMG is imperative to understanding whether/how much viral moderation of host function could influence geologic interpretation of the photosynthetic activity. In the methods it appears that work was undertaken to assess this abundance (line 135), but this data does not appear to be shown in the results.

Lines 333-338: As written, I find the relevance of the examples cited here to the results of this study difficult to follow.

Line 339- 342: References for similar findings in viral metagenomic datasets and cyanobacterial isolates are needed here, e.g. (Hurwitz et al., 2013, 2014; Lindell et al., 2004; Sullivan et al., 2003, 2005, 2006) .

Comment of section “Viruses encode carbon fixation genes” (lines 344-359). This section could be written more clearly. After some scrutiny and reading the paragraph that follows, I believe the arguments are: 1) PSB:GSB cell abundance is not a good proxy of photosynthetic activity; 2) Carbon isotope composition can be used to determine PSB:GSB photosynthetic production, but does not match recorded pigment concentrations; 3) By shutting down carbon fixation while increasing light reactions, viral AMGs alter carbon isotope fractionation, effecting predictions of PSB:GSB photosynthetic activity via this proxy. A summary such as this, in addition to the examples provided in the manuscript, may assist clarity here.

Line 377-378: “we found phage genomes encoding at least nine genes involved in sulfur

metabolism and relay system”. Include number of phages encoding with these genes, though again, what is really relevant and missing here is how abundant they are.

Lines 381 and 388: “deviate sulfur”. Is deviate is the right word here? Maybe ‘redirect’?

Line 389: “viral particle production may significantly modify the apparent sulfur fractionation”. Again, the significance of viral particle production to apparent sulfur fractionation will depend on how prevalent the modification of host machinery is in the environment, which may be predicted based on the abundance of the viruses that may have this ability. So the addition of this information is vital.

References

- Bryan, M. J., Burroughs, N. J., Spence, E. M., Clokie, M. R. J., Mann, N. H., & Bryan, S. J. (2008). Evidence for the intense exchange of MazG in marine cyanophages by horizontal gene transfer. *PLoS ONE*, 3(4), 1–12. <https://doi.org/10.1371/journal.pone.0002048>
- Forterre, P. (2012). The virocell concept and environmental microbiology. *The ISME Journal*, 7(2), 233–236. <https://doi.org/10.1038/ismej.2012.110>
- Fridman, S., Flores-Urbe, J., Larom, S., Alalouf, O., Liran, O., Yacoby, I., Salama, F., Bailleul, B., Rappaport, F., Ziv, T., Sharon, I., Cornejo-Castillo, F. M., Filosof, A., Dupont, C. L., Sánchez, P., Acinas, S. G., Rohwer, F. L., Lindell, D., & Béjà, O. (2017). A myovirus encoding both photosystem I and II proteins enhances cyclic electron flow in infected *Prochlorococcus* cells. *Nature Microbiology*, 2(10), 1350–1357. <https://doi.org/10.1038/s41564-017-0002-9>
- Hurwitz, B. L., Brum, J. R., & Sullivan, M. B. (2014). Depth-stratified functional and taxonomic niche specialization in the ‘core’ and ‘flexible’ Pacific Ocean Virome. *The ISME Journal*, 1–13. <https://doi.org/10.1038/ismej.2014.143>
- Hurwitz, B. L., Hallam, S. J., & Sullivan, M. B. (2013). Metabolic reprogramming by viruses in the sunlit and dark ocean. *Genome Biology*, 14(11), R123. <https://doi.org/10.1186/gb-2013-14-11-r123>
- Lindell, D., Jaffe, J. D., Coleman, M. L., Futschik, M. E., Axmann, I. M., Rector, T., Kettler, G., Sullivan, M. B., Steen, R., Hess, W. R., Church, G. M., & Chisholm, S. W. (2007). Genome-wide expression dynamics of a marine virus and host reveal features of co-evolution. *Nature*, 449(7158), 83–86. <https://doi.org/10.1038/nature06130>
- Lindell, D., Sullivan, M. B., Johnson, Z. I., Tolonen, A. C., Rohwer, F., & Chisholm, S. W. (2004). Transfer of photosynthesis genes to and from *Prochlorococcus* viruses. *Proceedings of the National Academy of Sciences of the United States of America*, 101(30), 11013–11018. <https://doi.org/10.1073/pnas.0401526101>
- Rihtman, B., Bowman-Grahl, S., Millard, A., Corrigan, R. M., Clokie, M. R. J., & Scanlan, D. J. (2019). Cyanophage MazG is a pyrophosphohydrolase but unable to hydrolyse magic spot nucleotides. *Environmental Microbiology Reports*, 11(3), 448–455. <https://doi.org/10.1111/1758-2229.12741>
- Shaffer, M., Borton, M. A., McGivern, B. B., Zayed, A. A., la Rosa, S. L. 0003 3527 8101, Solden, L. M., Liu, P., Narrowe, A. B., Rodríguez-Ramos, J., Bolduc, B., Gazitúa, M. C., Daly, R. A., Smith, G. J., Vik, D. R., Pope, P. B., Sullivan, M. B., Roux, S., & Wrighton, K. C. (2020). DRAM for distilling microbial metabolism to automate the curation of microbiome function. *Nucleic Acids Research*, 48(16), 8883–8900. <https://doi.org/10.1093/nar/gkaa621>
- Sullivan, M. B., Coleman, M. L., Weigele, P., Rohwer, F., & Chisholm, S. W. (2005). Three *Prochlorococcus* cyanophage genomes: Signature features and ecological interpretations. *PLoS Biology*, 3(5), 0790–0806. <https://doi.org/10.1371/journal.pbio.0030144>
- Sullivan, M. B., Lindell, D., Lee, J. A., Thompson, L. R., Bielawski, J. P., & Chisholm, S. W.

(2006). Prevalence and evolution of core photosystem II genes in marine cyanobacterial viruses and their hosts. *PLoS Biology*, 4(8), 1344–1357.

<https://doi.org/10.1371/journal.pbio.0040234>

Sullivan, M. B., Waterbury, J. B., & Chisholm, S. W. (2003). Cyanophages infecting the oceanic cyanobacterium *Prochlorococcus*. *Nature*, 424(6952), 1047–1051.

<https://doi.org/10.1038/nature02147>

Suttle, C. A. (2007). Marine viruses--major players in the global ecosystem. *Nature Reviews. Microbiology*, 5(10), 801–812. <https://doi.org/10.1038/nrmicro1750>

Reviewer #4 (Remarks to the Author):

The manuscript by Hesketh-Best et al describes several viral-encoded AMGs that may alter the metabolism of their host purple and green sulfur bacteria. The presence of these genes is hypothesized to affect metabolic markers used to reconstruct early oceans on Earth. I found this manuscript to be very straight forward, although I do suggest some additional explanation for a few of the ideas presented.

Line 45 – “GOE” is not yet defined as the abbreviation for “Great Oxygenation Event”.

Lines 189-193 & Lines 200-208 – Please cite where this data is presented in the manuscript.

Figure 1 – I cannot find where Figure 1B is cited within the manuscript.

Lines 32-33 - In abstract is the phrase “a pathway not yet described in these sulfur bacteria”. This was a very intriguing statement because viruses rarely encode all genes needed for a metabolic pathway. However, I read this section of the discussion related to sulfur metabolism (lines 373-397) several times but did not see discussion of this pathway not being present in the bacterial hosts. I think this should be stated in the discussion and the authors should expand upon their possible explanation for viruses having genes that may alter a metabolic pathway not present in their hosts.

Figure 4 - Part III of Figure 4 is not cited within the sulfur section of the discussion that I could see, and it is difficult to understand how the expression of these viral genes may affect amino acid synthesis based on looking at the figure.

Figure 4 - I also think Figure 4 could be expanded to illustrate how each of these scenarios would alter the metabolic markers used to interpret the early Earth record. This would help readers understand the central thesis of the manuscript in illustrated form as well as in the text form, which is I think what the authors intended.

General – The hypotheses presented in the manuscript are all based on putative viral genomes and the presence of AMGs within those genomes. While the authors are careful to state that this is a preliminary study, I do think it is warranted to include some statements of what should be done to actually demonstrate that these are active prophages, that the genes are expressed during infection, and that the genes actually alter the metabolic pathways as the authors suggest.

Reviewer: Dr Joanna Warwick-Dugdale

Manuscript entitled:

"Genomics of viruses infecting green and purple sulfur bacteria in two euxinic lakes"

By Dr Silveira and colleagues, submitted for publication in Communications Earth & Environment.

Summary

Overall, this study is an interesting investigation of important and novel putative bacteriophages infecting green sulfur bacteria (GSB) and purple sulfur bacteria (PSB) mined from: 1) Cellular, metagenomic long-read sequencing data obtained from the sediment and water column of two sulfidic and anoxic (euxinic) lakes; 2) publicly available GSB and PSB genomes. Having generated metagenome assembled bacterial genomes (MAGs) from their long-read data, the authors examine the composition of the bacterial communities derived from their euxinic lake samples, and the diversity of mined phage genomes. A combination of approaches utilising both the long-read data generated by the authors and publicly available GSB and PSB genomes are applied for prediction of phage hosts. Lastly, putative GSB- and PSB-infecting phage auxiliary metabolic genes (AMGs) are identified, and the authors hypothesise possible effects on host metabolism. In particular, the influence of viral AMGs is highlighted as having the potential to impact the biological signatures of GSB and PSB, and thus the interpretation in of their photosynthetic activity in the geologic record.

Major claims; comments

The major claims of the paper appear to consist of the following:

- 1) The putative viral genomes mined from the cellular metagenomic long-read data infect PSB and GSB hosts (where predicted).
- 2) Such viruses include AMGs that can redirect host metabolism and pigment production.
- 3) The presence of such AMGs may influence interpretation of the geologic record when it is based on GSB and PSB biological signatures.

For the most part this paper is clearly written, however the informatics applied require some further explanation/consideration. The authors make considerable efforts to associate hosts with the novel viruses they identify from euxinic lake sediment and water. Viral AMGs of interest that may redirect host metabolism in PSB and GSB are identified, however the level of curation required to ensure that they are encoded by viruses is not evident. There is concern regarding the generation of MAGs from low quality long reads, where the tools employed have been designed and tested using short reads (i.e. reads of similar length and similarly high accuracy), and without citations or prior tests to show

that the approach is valid. I also have criticisms the assessment of the possible significance of viral AMGs to host community function, and therefore interpretations of the geologic record, without: 1) knowledge of the abundance of such viruses in the environment; 2) consideration of the evidence (or lack thereof) for the functionality of the virus-encoded gene. Greater detail in the methods employed, and in some cases the logic behind choosing them (listed below), are needed to improve the reproducibility of the work. Some additional primary literature references for the methods employed (both 'wet-lab', and bioinformatic), and the effect of viral AMGs on host function, are currently missing.

Comments are listed by line and section, with the lines/sections that highlight issues of major concern in bold type.

Abstract and Introduction

Line 1: "euxinic". I would avoid use of this term in the title, as it is little used outside of the field (I would say "two sulfidic and anoxic lakes").

Line 35: "used as proxies to interpret". This sentence could be clearer; maybe "which are used as proxies for the interpretation of biogeochemical processes in early Earth Oceans".

Line 45: "pre-GOE". Acronym not specified in previous text (should be specified in line 38).

Line 46: "biotic factor controls". Viral infection is only *one* of the biotic factors that may contribute to the function of hosts (e.g. multicellular organisms are also biotic factors that affect host ecology).

Line 52: "as" should be "of".

Line 54: "are potential". It would be clearer to say that biomarkers etc *have* been used as proxies for basin depth and redox state.

Line 64 and Line 66: Better to include some primary literature here, as processes vital to arguments made later in the manuscript on the effect of viruses on host function are being introduced, e.g. (Forterre, 2012; Lindell et al., 2004, 2007; Suttle, 2007)

Line 69: "and sulfide oxidising phototrophs and". Reference for this? Also, I would exchange "and" for "thus/therefore".

Line 71: "concurrently". A better word might be simultaneously.

Line 72: "have high signatures": reads better as "High rates of horizontal gene transfer are suggested in by GSB genomes signatures".

Line 74 and 75: “Likewise [...] *pcyA*”): As later discussion includes the effect of phages on host ecology, it would be worth stating that the products of genes concerned with photosynthesis have been shown to actively alter rates of host photosynthesis in oxygenic hosts, and include the relevant references, e.g. (Fridman et al., 2017).

Methods

Line 87: Include depth of sulfidic zone and sample collection.

Lines 88-92: “Subsamples [...] Powersoil kit”): References where this method of DNA extraction has been successfully used in previous studies (ideally on water-column community) bacterial DNA are missing.

Line 92: “freeze core”): A brief description of this process, and how it has been modified from the reference provided, would improve understanding and reproducibility.

Line 93: “flow bench”): Flow hood?

Line 94: “An archive section...”): Not needed

Line 97-100: This is a very long sentence, and I found the meaning difficult to follow. If the 16S data is relevant, then some information on the methods employed to produce it, a representation of the results and analysis, and clearer statement of how they are important are needed here.

Line 104: “Metagenomic libraries were”): Worth naming the samples the libraries were generated from here.

Line 106: “1 mg of dsDNA”): Even for long-read sequencing, this seems like a *lot* of DNA to go into library preparation. Is this in total, or per sample? Could this amount of DNA have been recovered from 50 ml of lake water? Oxford Nanopore protocols call for assay of DNA quality prior to library preparation, as this is needed to calculate/approximate the molarity of the DNA required. Were any such assays obtained (for example, from an Agilent ‘TapeStation’)? This information will be needed for reproducibility.

Line 109: “using 1.8X Agencourt AMPure XP beads”): This is a higher volume of beads than specified in the manufacturer’s instructions (i.e., Oxford Nanopore protocol associated with the library preparation kit listed here). It should be clear where the manufacturer’s protocol has been modified, and it would be even better to include the reasons for the changes. I would be interested to know why this change was made. It seems that the majority of the long-read sequences obtained by the authors were low quality and <1000 bp long (line 183). Lower bead-volume to sample-volume ratios

are used to remove smaller fragments of DNA, and I wonder sticking to the original volumes specified in the protocol may help to improve read-length and quality.

Line 111: “FLOW-MINSP6”. Better to quote the ‘R’ number as this indicates the type of ONT chemistry employed (e.g. R9.41; R10.4); Also “loaded onto and”: not needed.

Line 118: “and quality”: Missing word ‘high’ (“and high quality”).

Line 130: State that VIBRANT was also used for identification of AMGs (shown in supplementary figure 1, but this step is important enough to the overall claims of paper to include here), and any references where this software has been used for the same purpose.

Line 132: State number of phages containing AMGs of interest that went on for further analysis here.

Line 135: “To analyse abundance and coverage”: where is abundance data for the phages (of interest) in the environment shown/discussed?

Comment on section “*Generation and quality control of MAGs*” (Lines 140-150): The tools employed by the authors (e.g. CONCOCT; metaBAT2) were designed and tested with mapping of short reads for binning (i.e. the reads being mapped are all similar length and similarly high accuracy). It appears that the assumption has been made that mappings made with long reads will work equally well, despite the varied read-length and lower accuracy of long reads. This assumption requires validation using a of mock dataset to show that the approach is not creating chimeric bins, or citation of prior work where such work has been undertaken. Additionally, based on supplementary figure 1, it's not clear why the low quality, short reads were mapped to the to the assembly for binning, which could potentially cause incorrect mapping. Why not just use the reads that are 1kb or greater in length and with a minimum Q-value of 9? Was this due to the scarcity of such reads in this dataset?

Lines 128-131: Of utmost importance for identifying viral AMGs, is that those which appear on the end of a contig should not be assumed as viral. The DRAM-v paper (Shaffer et al., 2020) explains why this is the case, and the cut-offs required to ensure that an AMG really is virally-encoded. If the authors have not used DRAM-V, then similar steps should have been used to curate their AMGs, neither of which currently appears in the text of the manuscript or in supplementary figure 1.

Lines: 166 176 (Section on *phage host prediction*: For parts I, II and II): Missing either references that have used the parameters stated, or some details on logic of choosing such.

Results

General comment after lines 181-182: "Line Blue (LB) sediment [...] "Poison Lake (PL) water."

Because sediment and water column environments and the microbial communities associated with them are so distinct, I think it would improve the readability and understanding of this manuscript if the samples, MAGs and putative viruses, were always referred to with the words 'sediment' or 'water', throughout.

Line 183: Very few reads of good quality and ≥ 1 kbp length. See comment on line 109.

Line 184: Interesting that so many more bacterial contigs were generated in LB sediment assemblies than PL water, despite that fact that fewer reads were generated from the sediment samples. Could this be due to the diversity of sediment bacteria compared to the water column? It would be good to know how many OTUs were generated from each sample type so that this idea could be considered.

Line 186: Figure 1 should be mentioned here as what it shows is being compared to the Supplementary Figures.

Line 219 and Table 1. "denoted in bold". Currently no values are shown in bold type.

General comment re. section: "*Diversity of PSB-infecting phages*": I think an additional supplementary table that summarises the following for each source of putative phage genomes (i.e. 'LB sediment'; 'PL water'; 'Bacterial MAGs from this study'; 'Publicly available PSB and GSB genomes') would be useful in clarifying the data: The total number of long reads generated and the number that were high quality (where appropriate); the number of high/med/low quality phages predicted; total number of hosts predicted and percentage that were GSB-PSB (via each method applied); the number of AMGs predicted. It could help convey an overall sense what the dataset comprises of.

Line 235: "metagenomic sequences,". Add "generated here," before for clarity.

Line 288: "AMG Abundances". The number of AMGs detected is just that – the number. In light of the authors' arguments on phage-AMG impact to interpretation of the geologic record, vital additional information is required on the *actual abundance* of those AMGs in the environment. How abundant are these AMGs/the viruses that encode these AMGs? If they are not encoded by the most abundant viruses, are they likely to create a signal strong enough to impact interpretation of the geologic record?

Line 300: "putative lytic phage": Add which sample this phage came from.

Discussion

Line 311: “viral genomes recovered from LB and PL metagenomes”. At the beginning of this section I think it would be more informative to state that these are *putative* viral genomes, and also that they come from cellular metagenomic samples rather than viral metagenomes (as viral metagenomic data is no longer rare).

Line 315: “proxy lakes”: The word proxy doesn’t make sense here – maybe ‘model lakes’?

Line 317: “(l) light harvesting molecule production”. Awkward phrasing; maybe “production of light harvesting molecules”

Line 321: “Viral predation”: “Viral Infection” would be better as it includes the influence of lysogens.

Line 326: “record as proxies”. Add the word ‘serve’ before ‘as proxies’, for clarity.

Line 327: “by physical and chemical bottom-up controls”. Include the appropriate references again here.

Line 328: “presented an alternative explanation”. Would ‘suggest’ be a better word than ‘presented’? Also, the word ‘alternative’ is not appropriate/needed.

Lines 328-332: I have two thoughts regarding the argument presented here.

Firstly, some acknowledgement of the caution needed in the assumption of viral AMGs having the same function as host genes is required here, as cases where this is not the case are in the literature. For example, based on host function, the phage encoded nucleotide pyrophosphohydrolase MazG was previously hypothesised to play a role in phage mediated regulation of the stringent response (i.e. host reaction to nutrient deprivation) (Bryan et al., 2008). However, examination of enzymatic activity indicated that instead, cyanophage MazG is likely to enable recycling of host DNA via the hydrolysis of deoxyribonucleotides (Rihtman et al., 2019). This is not to say that viral AMG function cannot be *predicted* from host gene function, but it should not be assumed. The argument for caution is especially pertinent when a function is extrapolated from a single gene in a synthetic pathway of many steps (as argued by the authors of the manuscript).

Secondly, the abundance of a viral AMG (or phage that encodes an AMG) is needed for evaluation of its possible importance to the ecosystem function. Here, knowing the abundance of the viral AMG is imperative to understanding whether/how much viral moderation of host function could influence geologic interpretation of the photosynthetic activity. In the methods it appears that work was undertaken to assess this abundance (line 135), but this data does not appear to be shown in the results.

Lines 333-338: As written, I find the relevance of the examples cited here to the results of this study difficult to follow.

Line 339- 342: References for similar findings in viral metagenomic datasets and cyanobacterial isolates are needed here, e.g. (Hurwitz et al., 2013, 2014; Lindell et al., 2004; Sullivan et al., 2003, 2005, 2006) .

Comment of section “*Viruses encode carbon fixation genes*” (lines 344-359). This section could be written more clearly. After some scrutiny and reading the paragraph that follows, I believe the arguments are: 1) PSB:GSB cell abundance is not a good proxy of photosynthetic activity; 2) Carbon isotope composition can be used to determine PSB:GSB photosynthetic production, but does not match recorded pigment concentrations; 3) By shutting down carbon fixation while increasing light reactions, viral AMGs alter carbon isotope fractionation, effecting predictions of PSB:GSB photosynthetic activity via this proxy. A summary such as this, in addition to the examples provided in the manuscript, may assist clarity here.

Line 377-378: “we found phage genomes encoding at least nine genes involved in sulfur metabolism and relay system”. Include number of phages encoding with these genes, though again, what is really relevant and missing here is how abundant they are.

Lines 381 and 388: “deviate sulfur”. Is deviate is the right word here? Maybe ‘redirect’?

Line 389: “viral particle production may significantly modify the apparent sulfur fractionation”. Again, the *significance* of viral particle production to apparent sulfur fractionation will depend on how prevalent the modification of host machinery is in the environment, which may be predicted based on the abundance of the viruses that may have this ability. So the addition of this information is vital.

References

- Bryan, M. J., Burroughs, N. J., Spence, E. M., Clokie, M. R. J., Mann, N. H., & Bryan, S. J. (2008). Evidence for the intense exchange of MazG in marine cyanophages by horizontal gene transfer. *PLoS ONE*, 3(4), 1–12. <https://doi.org/10.1371/journal.pone.0002048>
- Forterre, P. (2012). The virocell concept and environmental microbiology. *The ISME Journal*, 7(2), 233–236. <https://doi.org/10.1038/ismej.2012.110>
- Fridman, S., Flores-Uribe, J., Larom, S., Alalouf, O., Liran, O., Yacoby, I., Salama, F., Bailleul, B., Rappaport, F., Ziv, T., Sharon, I., Cornejo-Castillo, F. M., Filosof, A., Dupont, C. L., Sánchez, P., Acinas, S. G., Rohwer, F. L., Lindell, D., & Béjà, O. (2017). A myovirus encoding both

- photosystem I and II proteins enhances cyclic electron flow in infected *Prochlorococcus* cells. *Nature Microbiology*, 2(10), 1350–1357. <https://doi.org/10.1038/s41564-017-0002-9>
- Hurwitz, B. L., Brum, J. R., & Sullivan, M. B. (2014). Depth-stratified functional and taxonomic niche specialization in the ‘core’ and ‘flexible’ Pacific Ocean Virome. *The ISME Journal*, 1–13. <https://doi.org/10.1038/ismej.2014.143>
- Hurwitz, B. L., Hallam, S. J., & Sullivan, M. B. (2013). Metabolic reprogramming by viruses in the sunlit and dark ocean. *Genome Biology*, 14(11), R123. <https://doi.org/10.1186/gb-2013-14-11-r123>
- Lindell, D., Jaffe, J. D., Coleman, M. L., Futschik, M. E., Axmann, I. M., Rector, T., Kettler, G., Sullivan, M. B., Steen, R., Hess, W. R., Church, G. M., & Chisholm, S. W. (2007). Genome-wide expression dynamics of a marine virus and host reveal features of co-evolution. *Nature*, 449(7158), 83–86. <https://doi.org/10.1038/nature06130>
- Lindell, D., Sullivan, M. B., Johnson, Z. I., Tolonen, A. C., Rohwer, F., & Chisholm, S. W. (2004). Transfer of photosynthesis genes to and from *Prochlorococcus* viruses. *Proceedings of the National Academy of Sciences of the United States of America*, 101(30), 11013–11018. <https://doi.org/10.1073/pnas.0401526101>
- Rihtman, B., Bowman-Grahl, S., Millard, A., Corrigan, R. M., Clokie, M. R. J., & Scanlan, D. J. (2019). Cyanophage MazG is a pyrophosphohydrolase but unable to hydrolyse magic spot nucleotides. *Environmental Microbiology Reports*, 11(3), 448–455. <https://doi.org/10.1111/1758-2229.12741>
- Shaffer, M., Borton, M. A., McGivern, B. B., Zayed, A. A., la Rosa, S. L. 0003 3527 8101, Solden, L. M., Liu, P., Narrowe, A. B., Rodríguez-Ramos, J., Bolduc, B., Gazitúa, M. C., Daly, R. A., Smith, G. J., Vik, D. R., Pope, P. B., Sullivan, M. B., Roux, S., & Wrighton, K. C. (2020). DRAM for distilling microbial metabolism to automate the curation of microbiome function. *Nucleic Acids Research*, 48(16), 8883–8900. <https://doi.org/10.1093/nar/gkaa621>
- Sullivan, M. B., Coleman, M. L., Weigele, P., Rohwer, F., & Chisholm, S. W. (2005). Three *Prochlorococcus* cyanophage genomes: Signature features and ecological interpretations. *PLoS Biology*, 3(5), 0790–0806. <https://doi.org/10.1371/journal.pbio.0030144>
- Sullivan, M. B., Lindell, D., Lee, J. A., Thompson, L. R., Bielawski, J. P., & Chisholm, S. W. (2006). Prevalence and evolution of core photosystem II genes in marine cyanobacterial viruses and their hosts. *PLoS Biology*, 4(8), 1344–1357. <https://doi.org/10.1371/journal.pbio.0040234>
- Sullivan, M. B., Waterbury, J. B., & Chisholm, S. W. (2003). Cyanophages infecting the oceanic cyanobacterium *Prochlorococcus*. *Nature*, 424(6952), 1047–1051. <https://doi.org/10.1038/nature02147>
- Suttle, C. A. (2007). Marine viruses--major players in the global ecosystem. *Nature Reviews. Microbiology*, 5(10), 801–812. <https://doi.org/10.1038/nrmicro1750>

Response to Reviewers' comments:**Reviewer 1**

The authors report on the diversity and distribution of viral sequences in two euxinic lakes in the US by means of long-read sequencing of viral fraction (though this is my guess, as information about sample size fractionation isn't provided) in samples from lake water and sediments. In addition, authors analyzed available sequenced genomes of green and purple sulfur bacteria to identify the presence of sequences of viral origin. The manuscript is based on the descriptive analysis of 2 samples, which is the biggest limitation of this study. Overall, this is a clear, concise, and well-written manuscript. It describes intriguing findings and, surely, can significantly contribute to the improving of our understanding about the diversity of viruses infecting sulfur metabolizing bacteria. The methods used are appropriate and provide sufficient information to be repeatable. Results and discussion are centered round taxonomic classification of identified viral sequences, which is the major novelty of this study, and description of the potential role of viral AMGs, which is only speculative as no experimental evidence could be provided. Authors found several AMGs in viral contigs, potentially involved in carbon and sulfur metabolism of their hosts and based on these findings conclude that viruses might affect host physiology and biogeochemical cycling of sulfur. This conclusion is in agreement with other recent findings and suggestions provided by Berg et al. 2021 ISME 15: 1569-84 and Šulčius et al. 2021 Genes 12: 886.

Response: We thank the reviewer for their positive comments. The current study focuses on only two euxinic lakes. However, our finding that phages encoding the AMGs of interest in the genomes of PSB and GSB isolated from other lakes, sediments, freshwater creeks, and coastal seawater around the world and made available in the NCBI Database (Figure 2 and Supplementary File 1) suggests a broad distribution and significance of these viral genes.

We now provide additional information on sample preparation: bulk metagenomes (not size fractionated) were sequenced from the whole community (Lines 113-125), and viral genomes within the communities were identified computationally using benchmarked approaches (Lines 166-184).

Additional phylogenetic and structural analyses of the viral AMGs show significant amino acid sequence and structural conservation between the viral and hosts genes, which gives support to the idea of functional conservation (Figures 4 and Supplementary Figures 7-9).

Finally, we thank the reviewer for bringing to our attention the study by Šulčius et al 2021 that support the findings of our study. Their results were incorporated into this version of the manuscript (Discussion, Lines 461-464). The study by Berg and colleagues is explored in the introduction (Lines 88-89).

Reviewer 2

Summary

The manuscript uses samples collected from two euxinic lakes combined with metagenomics to show how viral infections affect green and purple sulfur bacteria (GSB and PSB, respectively) in these proposed early Earth analog environments. The manuscript states that the results show that viral infection of GSB and PSB interferes with photosynthesis, pigment production, and carbon and sulfur metabolism. According to the

manuscript, these results suggest that viral infection of GSB and PSB could impact the interpretation of their pigment biomarkers in the sedimentary record. In total, this was an interesting study, but major revisions are required prior to publication. These revisions relate to the motivation for the study and how the results impact the interpretation of GSB and PSB biomarkers in the rock record.

Response: We thank the reviewer for the comments that helped clarify the motivation and reach of the study. We now provide detailed information on the environmental observations that motivated the study in the introduction (described in detail below) and removed overstatements about the sedimentary record.

Major comments

1) The link between phototrophic sulfur bacteria and the GOE at the beginning of the introduction and conclusion feels forced and misaligned as a motivation for this study. For example, the oceans were likely ferruginous rather than euxinic prior to the GOE, the latter of which is more relevant for this study of euxinic lakes and GSB and PSB. The Johnston et al reference (ref #3) cited in the first sentence is specifically about anoxygenic phototrophic bacteria dominating during the Proterozoic (post GOE) rather than Archean (pre-GOE), which conflicts with the first sentence of the introduction where it is referenced.

Response: We agree with the reviewer that interpretations on the record pointed to the dominance of PSB and GSB on the Proterozoic oceans (post-GOE). In the revised version of the manuscript, we clarified that the the study of these lakes is motivated by their importance in understanding environments that predate the development of present oxygen levels in Earth's atmosphere and oceans (Lines 36-39).

Furthermore, the idea that GSB and PSB were significant primary producers prior to the GOE is not definitive as there is no direct evidence for it, and recent work suggests that Chlorobi may have postdated the GOE (e.g., Ward and Shih, 2022). The introduction and conclusion should be rewritten with a better justified motivation.

Response: Thank you for bringing to our attention recent literature on evidence of acquired capacity for anoxygenic photosynthesis via horizontal gene transfer sometime after the evolution of oxygenic Cyanobacteria. In the revised version of the manuscript, we motivate the study around the interpretations of euxinia in the geologic record in general, and not specifically pre-GOE (Lines 36-39). The first appearance of PSB and GSB biomarkers (okenane, chlorobactane, and isorenieratane) is from the 1.64 Ga Barney Creek Formation (Brocks et al., 2005, 2008). This implies that sulfur bacteria were also substantial part of aquatic microbial ecologies by this time.

2) The sampling subsection of the Methods section is lacking information about the study locations. For example, where are the lakes located, why were these two lakes chosen, in what ways are these lakes similar or different (e.g., water depth, size), etc? The discussion section is the first place where the lake locations are stated, whereas this information should occur in the sampling subsection. At minimum, the main manuscript text needs to contain some brief study site details, which is currently absent.

Response: We apologise for the oversight. We now describe in the methods the geographical coordinates for the lakes, the reason for choosing these locations, and several characteristics of the lakes (Lines 105-110, Supplementary Figure 1): "The research was conducted in two shallow (<16 m), sulfidic lakes: Lime Blue and Poison Lake, in Washington (WA), U.S.. The study sites are closed-basin lakes that lose water by

evaporation and seepage only and that receive water from direct precipitation, runoff, and catchment groundwater (Steinman et al., 2012). Undeveloped catchments, strong salinity gradients, and closed-basin configurations promote the prolonged periods of meromixis and benthic euxinia required by PSB and GSB, contributing to make these lakes ideal study sites.”

3) The discussion section contains multiple passages that are vague and need specifics and clarification. For example, lines 324-327 present a problem where the GSB and PSB biomarker records for photic zone euxinia “are not fully explained by physical and chemical bottom-up controls”. It is not clear what this passage is specifically referring to. Next, the text states that the metagenomic results present “an alternative explanation for this observation,” but it is not clear what the “observation” is that the new results are potentially explaining. In general, the text does not state the specific problem with the GSB and PSB biomarker record that the results address or how specifically the results impact how these biomarkers are interpreted in the geologic record. Rather, the manuscript only states that “this work will have important implications” for the GSB and PSB biomarker record (e.g., lines 330-333; 403-405). What are those implications specifically? How should readers incorporate these results into using and interpreting GSB and PSB biomarkers in deep time?

Response: We agree that improved reasoning is necessary. We now included in the introduction a detailed explanation about the decoupling between PSB and GSB distribution in modern lakes and their contribution to carbon fixation, pigment concentrations, and surficial isotope fractionation (Lines 60-63, 66-73). For example, according to Meyer et al. (2011), in Green Lake (NY), okenone (PSB product) is the major sulfide oxidizer biomarker in sediments. Yet, GSB is dominant in the water column. Other examples come from Lake Cadagno, in Switzerland, and Mahoney Lake, in Canada. Based on the data present in our manuscript, we hypothesize that viral-encoded metabolic genes may modulate the production of light-harvesting molecules (Figure 5). Simply put, viral infections could increase the production of okenone or modulate rates of carbon fixation. For the rock record, this means that these potential viral-mediated changes in biomarkers abundance need to be taken into account when using these pigments and isotopes as indicators of photic zone euxinia depth (Lines 417-420, 441-443, 478-482).

Minor comments

Line 38: I suggest using another word instead of culminating. Could simply provide approximate age of GOE.

Response: The statement has been removed.

Line 39: Ocean stratification is a result of density (temperature and salinity), so this is not the correct term here.

Response: This statement has been removed.

Line 50: Another word is needed to replace intercepts since this implies that the sulfide “stops” the sunlit layer.

Response: Changed to ‘reaches’ (line 49).

Line 54: Suggest “GSB and PSB pigments and their diagenetic products are biomarkers for...”

Response: Sentence rephrased (lines 53-55).

Line 55: It is not correct to state that GSB and PSB biomarkers are proxies for basin depth. Rather, they are used as an indicator for depth of intersection of photic zone and reduced sulfur, which is different from total depth of a basin or water column.

Response: Statement corrected (lines 55-57).

Figure 1: It would be useful to note in the figure and description which of the GSB and PSB produce the biomarkers mentioned in the text since not all GSB and PSB make the biomarker pigments.

Response: This has been noted as an additional sentence (Lines 249-252). Furthermore, we now indicate in Figure 1d the known producers of biomarkers.

Line 315: Suggest deleting “proxy” so the text only reads lakes, which is clearer.

Response: Removed as suggested.

Figure 4. It would be useful to illustrate in the figure (left cartoon) how viral infection affects okenone production.

Response: The conceptual figure (Figure 5 in revised manuscript) has been expanded as suggested.

Lines 333: “This hypothesis is consistent with ...” It does not seem that the hypothesis that this manuscript is proposing in lines 330-333 are either consistent or inconsistent with the previous work described in lines 333-339, rather more accurately these results are “in addition to” the previous work showing how pigment abundance may not accurately reflect the abundance of a specific taxon. Viral infection and HGT scramble this first order relationship.

Response: This statement has been removed.

Line 352: single GSB strain rather than single GSB?

Response: This statement was removed.

Line 381, 288: suggest using different word than deviate (e.g., divert).

Response: Changed as recommended (Line 465).

Line 386: S isotopes of sulfate and sulfide or similar rather than “sulfate and sulfide isotopes”

Response: Changed as recommended (Lines 450).

Suggest removing the PL and LB abbreviations since they are not commonly used and do not add clarity to the text.

Response: Changed as suggested throughout the revised manuscript to the full names of the sampling sites.

Typos: line 39, 395

Response: Both typos were corrected.

Reviewer 3

Summary

Overall, this study is an interesting investigation of important and novel putative bacteriophages infecting green sulfur bacteria (GSB) and purple sulfur bacteria (PSB) mined from: 1) Cellular, metagenomic long-read sequencing data obtained from the sediment and water column of two sulfidic and anoxic (euxinic) lakes; 2) publicly available GSB and PSB genomes. Having generated metagenome assembled bacterial genomes (MAGs) from their long read data, the authors examine the composition of the bacterial communities derived from their euxinic lake samples, and the diversity of mined phage genomes. A combination of approaches utilising both the long-read data generated by the authors and publicly available GSB and PSB genomes are applied for prediction of phage hosts. Lastly, putative GSB- and PSB-infecting phage auxiliary metabolic genes (AMGs) are identified, and the authors hypothesise possible effects on host metabolism. In particular, the influence of viral AMGs is highlighted as having the potential to impact the biological signatures of GSB and PSB, and thus the interpretation in of their photosynthetic activity in the geologic record.

Major claims; comments

The major claims of the paper appear to consist of the following:

- 1) The putative viral genomes mined from the cellular metagenomic long-read data infect PSB and GSB hosts (where predicted).
- 2) Such viruses include AMGs that can redirect host metabolism and pigment production.
- 3) The presence of such AMGs may influence interpretation of the geologic record when it is based on GSB and PSB biological signatures.

For the most part this paper is clearly written, however the informatics applied require some further explanation/consideration. The authors make considerable efforts to associate hosts with the novel viruses they identify from euxinic lake sediment and water. Viral AMGs of interest that may redirect host metabolism in PSB and GSB are identified, however the level of curation required to ensure that they are encoded by viruses is not evident. There is concern regarding the generation of MAGs from low quality long reads, where the tools employed have been designed and tested using short reads (i.e. reads of similar length and similarly high accuracy), and without citations or prior tests to show that the approach is valid. I also have criticisms the assessment of the possible significance of viral AMGs to host community function, and therefore interpretations of the geologic record, without: 1) knowledge of the abundance of such viruses in the environment; 2) consideration of the evidence (or lack thereof) for the functionality of the virus-encoded gene. Greater detail in the methods employed, and in some cases the logic behind choosing them (listed below), are needed to improve the reproducibility of the work. Some additional primary literature references for the methods employed (both 'wet-lab', and bioinformatic), and the effect of viral AMGs on host function, are currently missing.

Response: We thank the reviewer for the comments, which made the manuscript much clearer about the methods employed and added significant and exciting results (detailed revisions below).

Comments are listed by line and section, with the lines/sections that highlight issues of major concern in bold type (see attached Word Doc version of review for bold type).

Abstract and Introduction

Line 1: “euxinic”. I would avoid use of this term in the title, as it is little used outside of the field (I would say “two sulfidic and anoxic lakes”).

Response: The title was changed as recommended (Line 1-2)

Line 35: “used as proxies to interpret”. This sentence could be clearer; maybe “which are used as proxies for the interpretation of biogeochemical processes in early Earth Oceans”.

Response: Changed as recommended (Line 33)

Line 45: “pre-GOE”. Acronym not specified in previous text (should be specified in line 38).

Response: We have removed mention of the GOE.

Line 46: “biotic factor controls”. Viral infection is only one of the biotic factors that may contribute to the function of hosts (e.g. multicellular organisms are also biotic factors that affect host ecology).

Response: We modified this sentence to express that in this study we are specifically investigating one possible biological control, not the only one (Line 75-76).

Line 52: “as” should be “of”.

Response: Corrected (Line 449).

Line 54: “are potential”. It would be clearer to say that biomarkers etc have been used as proxies for basin depth and redox state.

Response: This statement has been removed.

Line 64 and Line 66: Better to include some primary literature here, as processes vital to arguments made later in the manuscript on the effect of viruses on host function are being introduced, e.g. (Forterre, 2012; Lindell et al., 2004, 2007; Suttle, 2007)

Response: References were added where appropriate (line 78).

Line 69: “and sulfide oxidising phototrophs and”. Reference for this? Also, I would exchange “and” for “thus/therefore”.

Response: The reference was added and the text was modified (Line 86).

Line 71: “concurrently”. A better word might be simultaneously.

Response: Changed as recommended (Line 89).

Line 72: “have high signatures”: reads better as “High rates of horizontal gene transfer are suggested in by GSB genomes signatures”.

Response: Changed as recommended (Lines 90-91).

Line 74 and 75: “Likewise [...] pcyA)”: As later discussion includes the effect of phages on host ecology, it would be worth stating that the products of genes concerned with photosynthesis have been shown to actively alter rates of host photosynthesis in oxygenic hosts, and include the relevant references, e.g. (Fridman et al., 2017).

Response: The text was rephrased (line 80-82).

Methods

Line 87: Include depth of sulfidic zone and sample collection.

Response: Sampling depth and lake vertical profiles are now described in Line 117 and supplementary Figure 1.

Lines 88-92: “Subsamples [...] Powersoil kit”: References where this method of DNA extraction has been successfully used in previous studies (ideally on water-column community) bacterial DNA are missing.

Response: Included as requested (Line 122)

Line 92: “freeze core”: A brief description of this process, and how it has been modified from the reference provided, would improve understanding and reproducibility.

Response: The freeze-coring procedure used as the same described in the reference provided (Line 123).

Line 93: “flow bench”: Flow hood?

Response: Corrected (line 123).

Line 94: “An archive section...”: Not needed

Response: Removed as recommended.

Line 97-100: This is a very long sentence, and I found the meaning difficult to follow. If the 16S data is relevant, then some information on the methods employed to produce it, a representation of the results and analysis, and clearer statement of how they are important are needed here.

Response: Mention of the 16S rRNA data has been removed as it was not relevant to this study, and this has shortened and improved the readability of this paragraph.

Line 104: “Metagenomic libraries were”: Worth naming the samples the libraries were generated from here.

Response: Changed as recommended (Line 128).

Line 106: “1 mg of dsDNA”: Even for long-read sequencing, this seems like a lot of DNA to go into library preparation. Is this in total, or per sample? Could this amount of DNA have been recovered from 50 ml of lake water? Oxford Nanopore protocols call for assay of DNA quality prior to library preparation, as this is needed to calculate/approximate the molarity of the DNA required. Were any such assays obtained (for example, from an Agilent ‘TapeStation’)? This information will be needed for reproducibility.

Response: We apologise for this mistake, it should have been 1 µg. This has been corrected and the methods for quantifying the DNA added to the methods (line 131).

Line 109: “using 1X Agencourt AMPure XP beads”: This is a higher volume of beads than specified in the manufacturer’s instructions (i.e., Oxford Nanopore protocol associated with the library preparation kit listed here). It should be clear where the manufacturer’s protocol has been modified, and it would be even better to include the reasons for the changes. I

would be interested to know why this change was made. It seems that the majority of the long-read sequences obtained by the authors were low quality and <1000 bp long (line 183). Lower bead-volume to sample-volume ratios are used to remove smaller fragments of DNA, and I wonder sticking to the original volumes specified in the protocol may help to improve read-length and quality.

Response: We apologise for this inaccuracy, which has now been corrected with the exact volume of AMPure XP beads that was added to the reaction mixture after the End-prep step (60 ul), as required in the ONT protocol (Line 134).

Line 111: "FLOW-MINSP6". Better to quote the 'R' number as this indicates the type of ONT chemistry employed (e.g. R9.41; R10.4); Also "loaded onto and": not needed.

Response: Added the R chemistry number (R9) as recommended (Line 137).

Line 118: "and quality": Missing word 'high' ("and high quality").

Response: Changed as recommended (Line 145).

Line 130: State that VIBRANT was also used for identification of AMGs (shown in supplementary figure 1, but this step is important enough to the overall claims of paper to include here), and any references where this software has been used for the same purpose.

Response: We have clarified that AMGs were acquired from VIBRANTs output. VIBRANT identifies viral auxiliary metabolic genes (AMGs) through HMM comparisons with three databases: Kyoto Encyclopedia of Genes and Genomes (KEGG) KoFam (March 2019 release), Pfam (v32), and Virus Orthologous Groups (VOG) (release 94, vogdb.org). VIBRANT utilises a manually curated collection of viral AMGs from KEGG annotations falling under the "metabolic pathways" category as well as "sulfur relay system" were considered, reduced to contain only profiles likely to annotate viruses of interest. (Lines 204-215).

Line 132: State number of phages containing AMGs of interest that went on for further analysis here.

Response: This detail has been added to the results (Lines 309-311): "Poison Lake and Lime Blue phages encoded 52 and 96 AMGs, respectively, representing 153 distinct KEGG pathways".

Line 135: "To analyse abundance and coverage": where is abundance data for the phages (of interest) in the environment shown/discussed?

Response: Phage fractional abundances can be found in the Supplementary File 2, and are shown in Figures 3c and 3d.

Comment on section "Generation and quality control of MAGs" (Lines 140-150): The tools employed by the authors (e.g. CONCOCT; metaBAT2) were designed and tested with mapping of short reads for binning (i.e. the reads being mapped are all similar length and similarly high accuracy). It appears that the assumption has been made that mappings made with long reads will work equally well, despite the varied read-length and lower accuracy of long reads. This assumption requires validation using a of mock dataset to show that the approach is not creating chimeric bins, or citation of prior work where such work has been undertaken. Additionally, based on supplementary figure 1, it's not clear why the low quality, short reads were mapped to the to the assembly for binning, which could

potentially cause incorrect mapping. Why not just use the reads that are 1kb or greater in length and with a minimum Q-value of 9? Was this due to the scarcity of such reads in this dataset?

Response: We understand the reviewer's concern regarding MAG using only a Nanopore dataset. However, similar pipelines and binning software have recently been utilised and benchmarked on the ZymoBIOMICS gut microbiome standard (Liu, Yang and Deng, 2022), resulting in the recovery of high-quality MAGs in a pipeline named *NanoPhase*. Other works show similar binning success using Nanopore-only and combined Nanopore-Illumina data (Singleton *et al.*, 2021).

To address the reviewer's concern, we performed additional binning with the benchmarked pipeline *NanoPhase* (Liu, Yang and Deng, 2022) and a specialised long read binner (LRBinner; Wickramarachchi and Lin, 2022). This resulted in the same bins being generated as from the initial method presented in this manuscript. Particularly, a bin taxonomically classified by GTDB-tK as *Thiohalocapsa* sp. and identified as a putative phage host to a virus containing an AMG of interest, was binned by all three binning strategies, with a bin identity between them of >95% ANI (see below highlighted in purple). The results of this additional binning has been presented in a MASH clustering tree (Supplementary Fig. 3/see below), denoting the original bins (CONCOCT+MetaBat2+MaxBIN2), and the bins resolved utilised the *NanoPhase* pipeline, and LRBinner. The binning strategies presented here recovered metagenomic bins that satisfy the minimum information for metagenome assembled genomes (MiMAG) requirements for contamination and completion as outlined in previous work (Bowers *et al.*, 2017).

Lines 128-131: Of utmost importance for identifying viral AMG, is that those which appear on the end of a contig should not be assumed as viral. The DRAM-v paper (Shaffer et al., 2020) explains why this is the case, and the cut-offs required to ensure that an AMG really is virally encoded. If the authors have not used DRAM-V, then similar steps should have been used to curate their AMG, neither of which currently appears in the text of the manuscript or in supplementary figure 1.

Response: None of the AMGs of interest are located at the end of a contig (Figure 3B), and all of them were present in contigs with viral hallmark genes, including the Major Capsid Protein. We additionally employed protein phylogeny and structural analyses comparing viral and host proteins (Methods Lines 216-227, Figure 4, and Supplementary Figures 7-9). Both phylogenetic and structural analyses provided strong support for functional conservation. This manual approach is more conservative than that employed by DRAM-v, which uses BLAST annotation using the viral NCBI RefSeq database and a very small database of experimentally tested AMGs (12 genes) plus AMGs that were bioinformatically predicted by amino acid sequence identity. Finally, DRAM-v also imposes a substantial memory burden (>500GB RAM), which were unfortunately beyond the capabilities of the computing system utilised for this study.

Lines: 166 176 (Section on phage host prediction: For parts I, II and II): Missing either references that have used the parameters stated, or some details on logic of choosing such.

Response: References added to Lines 193, 199, 201.

Results

General comment after lines 181-182: “Line Blue (LB) sediment [...] “Poison Lake (PL) water.” Because sediment and water column environments and the microbial communities associated with them are so distinct, I think it would improve the readability and understanding of this manuscript if the samples, MAGs and putative viruses, were always referred to with the words ‘sediment’ or ‘water’, throughout.

Response: We have emphasised throughout the revised manuscript that Lime Blue is a sediment metagenome and that Poison Lake is a water column metagenome.

Line 183: Very few reads of good quality and ≥ 1 kbp length. See comment on line 109.

Response: The bead cleanup step followed the ONT instructions and was unlikely the cause of low-quality reads.

Line 184: Interesting that so many more bacterial contigs were generated in LB sediment assemblies than PL water, despite that fact that fewer reads were generated from the sediment samples. Could this be due to the diversity of sediment bacteria compared to the water column? It would be good to know how many OTUs were generated from each sample type so that this idea could be considered.

Response: Supplementary table 1 has been added and contains a summary of the data generated, containing the number of distinct Phyla and Species recovered from the two samples. Indeed, Poison Lake water seems to be more diverse, which could be the reason for poor assembly compared to Lime Blue sediment.

Line 186: Figure 1 should be mentioned here as what it shows is being compared to the Supplementary Figures.

Response: This sentence has now been rephrased and we focus more on the bin diversity in Lines 255-267.

Line 219 and Table 1. “denoted in bold”. Currently no values are shown in bold type. General comment re. section: “Diversity of PSB-infecting phages”: I think an additional supplementary table that summarises the following for each source of putative phage genomes (i.e. ‘LB sediment’; ‘PL water’; ‘Bacterial MAGs from this study’; ‘Publicly available PSB and GSB genomes’) would be useful in clarifying the data: The total number of long reads generated and the number that were high quality (where appropriate); the number of high/med/low quality phages predicted; total number of hosts predicted and percentage that were GSB-PSB (via each method applied); the number of AMGs predicted. It could help convey an overall sense what the dataset comprises of.

Response: We now used symbols (not bold) to label items in Table 1. We also provide a Supplementary Table 1 containing the information requested.

Line 235: “metagenomic sequences,”. Add “generated here,” before for clarity.

Response: This sentence has been rephrased as follows: “VIBRANT identified 2,742 putative phage genomes from Lime Blue contigs (100 medium-quality genomes, 24 high-quality, and two complete circular genomes) and 5,806 from Poison Lake metagenomic reads, all of which were low-quality phage genome fragments.” (Line 270).

Line 288: “AMG Abundances”. The number of AMGs detected is just that – the number. In light of the authors’ arguments on phage-AMG impact to interpretation of the geologic record, vital additional information is required on the actual abundance of those AMGs in the environment. How abundant are these AMGs/the viruses that encode these AMGs? If they are not encoded by the most abundant viruses, are they likely to create a signal strong enough to impact interpretation of the geologic record?

Response: A rank abundance curve of now shows the abundance of the AMG-encoding phages in the community (Figures 3c and d). High abundance phages (between ranks 70 and 400) encode the AMGs of interest (eg. *psbA*, *crtF*, *cysH*, *G6PD*; Lines 329, 370, 335, 375).

Line 300: “putative lytic phage”: Add which sample this phage came from.

Response: We have clarified that the phage was derived from Lime Blue sample (Line 328).

Discussion

Line 311: “viral genomes recovered from LB and PL metagenomes”. At the beginning of this section I think it would be more informative to state that these are putative viral genomes, and also that they come from cellular metagenomic samples rather than viral metagenomes (as viral metagenomic data is no longer rare).

Response: We now use “putative viral genomes” as recommended throughout the text.

Line 315: “proxy lakes”: The word proxy doesn’t make sense here – maybe ‘model lakes’?

Response: Sentence has been changed, and the term ‘proxy lakes’ removed.

Line 317: “(l) light harvesting molecule production”. Awkward phrasing; maybe “production of light harvesting molecules”

Response: Changed to ‘the synthesis of light-harvesting molecules’ (Line 384).

Line 321: “Viral predation”: “Viral Infection” would be better as it includes the influence of lysogens.

Response: Changed as recommended (Line 394).

Line 326: “record as proxies”. Add the word ‘serve’ before ‘as proxies’, for clarity.

Response: Changed to ‘used as proxies’ (Lines 33 and 390).

Line 327: “by physical and chemical bottom-up controls”. Include the appropriate references again here.

Response: This sentence has been removed in the revised manuscript.

Line 328: “presented an alternative explanation”. Would ‘suggest’ be a better word than ‘presented’? Also, the word ‘alternative’ is not appropriate/needed.

Response: This sentence was modified as follows: “Viral-mediated changes in biomarker abundance would need to be considered when using pigment biomarkers as indicators of photic zone euxinia depth.” (Lines 419-420).

Lines 328-332: I have two thoughts regarding the argument presented here.

Firstly, some acknowledgement of the caution needed in the assumption of viral AMGs having the same function as host genes is required here, as cases where this is not the case are in the literature. For example, based on host function, the phage encoded nucleotide pyrophosphohydrolase MazG was previously hypothesised to play a role in phage mediated regulation of the stringent response (i.e. host reaction to nutrient deprivation) (Bryan et al., 2008). However, examination of enzymatic activity indicated that instead, cyanophage MazG is likely to enable recycling of host DNA via the hydrolysis of deoxyribonucleotides (Rihtman et al., 2019). This is not to say that viral AMG function cannot be predicted from host gene function, but it should not be assumed. The argument for caution is especially pertinent when a function is extrapolated from a single gene in a synthetic pathway of many steps (as argued by the authors of the manuscript).

Response: As described above, our structural analysis suggests functional conservation. Yet, to use caution, as suggested by the reviewer, we added a limitations section where we state that functional studies are necessary (Lines 484-497).

Secondly, the abundance of a viral AMG (or phage that encodes an AMG) is needed for evaluation of its possible importance to the ecosystem function. Here, knowing the abundance of the viral AMG is imperative to understanding whether/how much viral moderation of host function could influence geologic interpretation of the photosynthetic activity. In the methods it appears that work was undertaken to assess this abundance (line 135), but this data does not appear to be shown in the results.

Response: Figure 3 now shows the phage abundances, revealing that phages encoding the AMGs of interest are among the most abundant viruses in these communities.

Lines 333-338: As written, I find the relevance of the examples cited here to the results of this study difficult to follow.

Response: This section has been rephased as follows: “Previous work showed that horizontal gene transfer in Lake Banyoles (Spain) results in the unexpected synthesis of photosynthetic pigments (bacteriochlorophyll e and isorenieratene) by green-pigmented GSB, *Chlorobium luteolum*, a bacterium that usually synthesizes bacteriochlorophyll c²⁵. This gene transfer event offered a fitness advantage to *C. luteolum* over brown-pigmented GSB by the expansion of its photo-adaptation range to a deeper photic zone. This example of Lake Banyoles is evidence that exogenous genes acquired laterally may affect pigment production, supporting the idea that phage genes in the Lime Blue may affect pigment synthesis in PSB.” (Lines 403-410).

Line 339- 342: References for similar findings in viral metagenomic datasets and cyanobacterial isolates are needed here, e.g. (Hurwitz et al., 2013, 2014; Lindell et al., 2004; Sullivan et al., 2003, 2005, 2006) .

Response: These references were added as recommended (Line 434).

Comment of section “Viruses encode carbon fixation genes” (lines 344-359). This section could be written more clearly. After some scrutiny and reading the paragraph that follows, I believe the arguments are: 1) PSB:GSB cell abundance is not a good proxy of photosynthetic activity; 2) Carbon isotope composition can be used to determine PSB:GSB photosynthetic production, but does not match recorded pigment concentrations; 3) By shutting down carbon fixation while increasing light reactions, viral AMGs alter carbon isotope

fractionation, effecting predictions of PSB:GSB photosynthetic activity via this proxy. A summary such as this, in addition to the examples provided in the manuscript, may assist clarity here.

Response: This section was rephrased as recommended (Lines 423-443).

Line 377-378: “we found phage genomes encoding at least nine genes involved in sulfur metabolism and relay system”. Include number of phages encoding with these genes, though again, what is really relevant and missing here is how abundant they are.

Response: Figure 3 now has 2 additional panels that indicate the abundance of the phages containing the AMGs of interest.

Lines 381 and 388: “deviate sulfur”. Is deviate is the right word here? Maybe ‘redirect’?

Response: Changed to ‘divert’ (Line 465).

Line 389: “viral particle production may significantly modify the apparent sulfur fractionation”. Again, the significance of viral particle production to apparent sulfur fractionation will depend on how prevalent the modification of host machinery is in the environment, which may be predicted based on the abundance of the viruses that may have this ability. So the addition of this information is vital.

Response: The abundance of these viruses is now shown in Figures 3c and d.

References

- Bryan, M. J., Burroughs, N. J., Spence, E. M., Clokie, M. R. J., Mann, N. H., & Bryan, S. J. (2008). Evidence for the intense exchange of MazG in marine cyanophages by horizontal gene transfer. *PLoS ONE*, 3(4), 1–12. <https://doi.org/10.1371/journal.pone.0002048>
- Forterre, P. (2012). The virocell concept and environmental microbiology. *The ISME Journal*, 7(2), 233–236. <https://doi.org/10.1038/ismej.2012.110>
- Fridman, S., Flores-Urbe, J., Larom, S., Alalouf, O., Liran, O., Yacoby, I., Salama, F., Bailleul, B., Rappaport, F., Ziv, T., Sharon, I., Cornejo-Castillo, F. M., Philosof, A., Dupont, C. L., Sánchez, P., Acinas, S. G., Rohwer, F. L., Lindell, D., & Béjà, O. (2017). A myovirus encoding both photosystem I and II proteins enhances cyclic electron flow in infected *Prochlorococcus* cells. *Nature Microbiology*, 2(10), 1350–1357. <https://doi.org/10.1038/s41564-017-0002-9>
- Hurwitz, B. L., Brum, J. R., & Sullivan, M. B. (2014). Depth-stratified functional and taxonomic niche specialization in the ‘core’ and ‘flexible’ Pacific Ocean Virome. *The ISME Journal*, 1–13. <https://doi.org/10.1038/ismej.2014.143>
- Hurwitz, B. L., Hallam, S. J., & Sullivan, M. B. (2013). Metabolic reprogramming by viruses in the sunlit and dark ocean. *Genome Biology*, 14(11), R123. <https://doi.org/10.1186/gb-2013-14-11-r123>
- Lindell, D., Jaffe, J. D., Coleman, M. L., Futschik, M. E., Axmann, I. M., Rector, T., Kettler, G., Sullivan, M. B., Steen, R., Hess, W. R., Church, G. M., & Chisholm, S. W. (2007). Genome-wide expression dynamics of a marine virus and host reveal features of co-evolution. *Nature*, 449(7158), 83–86. <https://doi.org/10.1038/nature06130>
- Lindell, D., Sullivan, M. B., Johnson, Z. I., Tolonen, A. C., Rohwer, F., & Chisholm, S. W. (2004). Transfer of photosynthesis genes to and from *Prochlorococcus* viruses. *Proceedings of the National Academy of Sciences of the United States of America*, 101(30), 11013–11018. <https://doi.org/10.1073/pnas.0401526101>
- Rihtman, B., Bowman-Grahl, S., Millard, A., Corrigan, R. M., Clokie, M. R. J., & Scanlan, D. J. (2019). Cyanophage MazG is a pyrophosphohydrolase but unable to hydrolyse magic spot

nucleotides. *Environmental Microbiology Reports*, 11(3), 448–455. <https://doi.org/10.1111/1758-2229.12741>

Shaffer, M., Borton, M. A., McGivern, B. B., Zayed, A. A., la Rosa, S. L. 0003 3527 8101, Solden, L. M., Liu, P., Narrowe, A. B., Rodríguez-Ramos, J., Bolduc, B., Gazitúa, M. C., Daly, R. A., Smith, G. J., Vik, D. R., Pope, P. B., Sullivan, M. B., Roux, S., & Wrighton, K. C. (2020). DRAM for distilling microbial metabolism to automate the curation of microbiome function. *Nucleic Acids Research*, 48(16), 8883–8900. <https://doi.org/10.1093/nar/gkaa621>

Sullivan, M. B., Coleman, M. L., Weigele, P., Rohwer, F., & Chisholm, S. W. (2005). Three *Prochlorococcus cyanophage* genomes: Signature features and ecological interpretations. *PLoS Biology*, 3(5), 0790–0806. <https://doi.org/10.1371/journal.pbio.0030144>

Sullivan, M. B., Lindell, D., Lee, J. A., Thompson, L. R., Bielawski, J. P., & Chisholm, S. W. (2006). Prevalence and evolution of core photosystem II genes in marine cyanobacterial viruses and their hosts. *PLoS Biology*, 4(8), 1344–1357. <https://doi.org/10.1371/journal.pbio.0040234>

Sullivan, M. B., Waterbury, J. B., & Chisholm, S. W. (2003). Cyanophages infecting the oceanic cyanobacterium *Prochlorococcus*. *Nature*, 424(6952), 1047–1051. <https://doi.org/10.1038/nature02147>

Suttle, C. A. (2007). Marine viruses--major players in the global ecosystem. *Nature Reviews. Microbiology*, 5(10), 801–812. <https://doi.org/10.1038/nrmicro1750>

Reviewer 4

The manuscript by Hesketh-Best et al describes several viral-encoded AMGs that may alter the metabolism of their host purple and green sulfur bacteria. The presence of these genes is hypothesized to affect metabolic markers used to reconstruct early oceans on Earth. I found this manuscript to be very straight forward, although I do suggest some additional explanation for a few of the ideas presented.

Line 45 – “GOE” is not yet defined as the abbreviation for “Great Oxygenation Event”.

Response: Mention of the GEO has been removed from the manuscript. Instead, we motivate our study in the context of present levels of oxygen (Line 37).

Lines 189-193 & Lines 200-208 – Please cite where this data is presented in the manuscript.

Response: These results are shown in figures 1c and d, and have been added into the text (Lines 238, 240, 244).

Figure 1 – I cannot find where Figure 1B is cited within the manuscript.

Response: Figure 1d is cited in the revised manuscript in line 240.

Lines 32-33 - In abstract is the phrase “a pathway not yet described in these sulfur bacteria”. This was a very intriguing statement because viruses rarely encode all genes needed for a metabolic pathway. However, I read this section of the discussion related to sulfur metabolism (lines 373-397) several times but did not see discussion of this pathway not being present in the bacterial hosts. I think this should be stated in the discussion and the authors should expand upon their possible explanation for viruses having genes that may alter a metabolic pathway not present in their hosts.

Response: We agree with the reviewer that this was not well described, so we removed this sentence from the abstract due to word limits and we expanded the proposal of how these genes may affect host metabolism of sulfur in the discussion (Lines 465-482).

Figure 4 - Part III of Figure 4 is not cited within the sulfur section of the discussion that I could see, and it is difficult to understand how the expression of these viral genes may affect amino acid synthesis based on looking at the figure.

Response: The conceptual figure (Figure 5 in revised manuscript) has been expanded to take into account the reviewers' concerns.

Figure 4 - I also think Figure 4 could be expanded to illustrate how each of these scenarios would alter the metabolic markers used to interpret the early Earth record. This would help readers understand the central thesis of the manuscript in illustrated form as well as in the text form, which is I think what the authors intended.

Response: The conceptual figure (Figure 5 in revised manuscript) has been expanded by inclusion of predicted changes in biosignatures.

General – The hypotheses presented in the manuscript are all based on putative viral genomes and the presence of AMGs within those genomes. While the authors are careful to state that this is a preliminary study, I do think it is warranted to include some statements of what should be done to actually demonstrate that these are active prophages, that the genes are expressed during infection, and that the genes actually alter the metabolic pathways as the authors suggest.

Response: A subsection entitled “*Limitations and future directions*” has been added to address the reviewers concerns (Lines 484-497).

10th Mar 23

Dear Dr Silveira,

Your manuscript titled "Genomics of viruses infecting green and purple sulfur bacteria in two anoxic and sulfidic lakes" has now been seen by our reviewers, whose comments appear below. In light of their advice I am delighted to say that we are happy, in principle, to publish a suitably revised version in Communications Earth & Environment under the open access CC BY license (Creative Commons Attribution v4.0 International License).

We therefore invite you to revise your paper one last time to address the remaining concerns of our reviewers, which will require removing the reference to Early Earth analogs from the abstract and appropriately toning down/caveating/removing similar claims elsewhere in the manuscript. At the same time we ask that you edit your manuscript to comply with our format requirements and to maximise the accessibility and therefore the impact of your work.

EDITORIAL REQUESTS:

*****Please take care to match our formatting and policy requirements. We will check revised manuscript and return manuscripts that do not comply. Such requests will lead to delays. *****

SUBMISSION INFORMATION:

OPEN ACCESS:

Communications Earth & Environment is a fully open access journal. Articles are made freely accessible on publication under a [CC BY license](http://creativecommons.org/licenses/by/4.0) (Creative Commons Attribution 4.0 International License). This license allows maximum dissemination and re-use of open access materials and is preferred by many research funding bodies.

For further information about article processing charges, open access funding, and advice and support from Nature Research, please visit <https://www.nature.com/commsenv/article-processing-charges>

At acceptance, you will be provided with instructions for completing this CC BY license on behalf of all authors. This grants us the necessary permissions to publish your paper. Additionally, you will be asked to declare that all required third party permissions have been obtained, and to provide billing information in order to pay the article-processing charge (APC).

[link redacted]

Best regards,

Clare Davis, PhD
Senior Editor
Communications Earth & Environment

www.nature.com/commsenv/
@CommsEarth

REVIEWERS' COMMENTS:

Reviewer #2 (Remarks to the Author):

The revised version of the manuscript, which has improved, addresses many of my previous comments. However, I recommend additional revisions prior to publication for the reasons listed below.

1) Related to my previous review, I recommend the authors reconsider the Early Earth motivation which is throughout the revised abstract, introduction, and conclusion. The manuscript would be better served by generalizing the motivation of this study to better understand what governs the abundance and distribution of biomarkers of phototrophic sulfur bacteria that serve as indicators of photic zone euxinia in the rock record. In fact, these biomarkers are far more frequently detected and reported in the Phanerozoic (e.g., mass extinctions and oceanic anoxic events) than during “early Earth” (e.g., French et al., 2015). As a result, the findings of this study are potentially equally or even more relevant to the Phanerozoic biomarker record than the Precambrian biomarker record. Therefore, statements such as those contained in Lines 94-95 and 504-505 better encompass the motivation and potential impact of the study. Statements like “anoxic and sulfidic lakes, analogs of early Earth oceans” in line 25 can be simply generalized to “anoxic and sulfidic lakes, analogs of euxinic oceans in the geologic past”. As a result, I recommend that the Early Earth parts of the manuscript be rewritten. Doing so would more accurately capture the motivation and outcome of the study, which has broader impacts beyond Early Earth.

2) Some passages need to be rewritten for clarity, specificity, accuracy. I recommend that the authors revisit the manuscript again before publication to improve the text. I have listed some examples here.

- a. Lines 33-34. This vague statement could be revised to a more specific statement such as “modulate production of metabolic markers of phototrophic sulfur bacteria that are used to identify photic zone euxinia in the geologic past”
- b. Lines 55-57. This sentence needs rewriting since it suggests that chlorobactene, isorenieratene, and okenone are the diagenetic products, which is inaccurate as these are the biological precursor carotenoid pigments.
- c. Lines 94-95: suggest something like “phage regulation of phototrophic sulfur bacteria pigment synthesis may affect the abundance and distribution of the GSB and PSB biomarkers that are used as indicators of PZE in the rock record” as it is more specifically worded. Otherwise, aspects of this sentence are vague.
- d. Line 124: revise to “was extracted for DNA” since pigment extraction is also possible and could be relevant to this manuscript.
- e. Lines 249-252: rewrite the sentence to make it more accurate and specific. “Known producers of diagenetic products” would be more accurately written as “Known producers of okenone, which is the biological precursor of the biomarker okenane,...” or “...which is the precursor of the diagenetic product okenane”.
- f. Line 389- rewrite since the sedimentary products are not light-harvesting necessarily, rather they are diagenetically related to light harvesting pigments.
- g. Line 392: suggest revising to “...pigments in sediments” since sedimentary record applies more ancient timescales and the cited studies are from modern sediments.
- h. Lines 451-453: Needs revision. The comma is confusing or a word is missing.
- i. Lines 480-482: Rewrite and simplify wording
- j. Lines 504-505: Simplify. Suggest “Our observations suggest that viral infections could impact biosignatures of phototrophic sulfur bacteria in the sedimentary record.”

3) Would it be reasonable to expand on lines 419-420 based on the results? For example, would it be reasonable to suggest that viral infection could lead to higher okenone abundances such that okenane abundances in the sedimentary record could overestimate PSB abundances and therefore the shallowness of PZE? Would the presented result and results from other systems, such as Green Lake, support this more specific statement?

Reviewer #3 (Remarks to the Author):

I have enjoyed reading the revised version of this manuscript. It is clear that the authors have gone to some lengths to address my questions regarding the provenance and abundance of the viral AMGs identified, and the validity of the MAGs generated. The modifications made to the methods section improves clarity and reproducibility, and the updated version of the discussion better suits the scope of the results and is far easier to follow. I am happy to conclude that the authors have addressed my previous concerns.

In addition, it is clear that the authors have thoroughly addressed Reviewer #4's concerns. The

discussion on possible alterations to host sulfur metabolism and the revisions to Figure 4 (now Figure 5) and do an excellent job in clarifying the authors' ideas of how different viral AMGs could alter metabolic markers used to interpret the geologic record (Reviewer #4's chief concern). I do think this MS is looking very solid now.

REVIEWERS' COMMENTS AND AUTHOR RESPONSES:

Reviewer #2 (Remarks to the Author):

The revised version of the manuscript, which has improved, addresses many of my previous comments. However, I recommend additional revisions prior to publication for the reasons listed below.

1) Related to my previous review, I recommend the authors reconsider the Early Earth motivation which is throughout the revised abstract, introduction, and conclusion. The manuscript would be better served by generalizing the motivation of this study to better understand what governs the abundance and distribution of biomarkers of phototrophic sulfur bacteria that serve as indicators of photic zone euxinia in the rock record. In fact, these biomarkers are far more frequently detected and reported in the Phanerozoic (e.g., mass extinctions and oceanic anoxic events) than during “early Earth” (e.g., French et al., 2015). As a result, the findings of this study are potentially equally or even more relevant to the Phanerozoic biomarker record than the Precambrian biomarker record. Therefore, statements such as those contained in Lines 94-95 and 504-505 better encompass the motivation and potential impact of the study. Statements like “anoxic and sulfidic lakes, analogs of early Earth oceans” in line 25 can be simply generalized to “anoxic and sulfidic lakes, analogs of euxinic oceans in the geologic past”. As a result, I recommend that the Early Earth parts of the manuscript be rewritten. Doing so would more accurately capture the motivation and outcome of the study, which has broader impacts beyond Early Earth.

Response: Response: We thank the reviewer for this insight. We agree and removed the passages that referred to early Earth, expressing motivation based on the geologic past in general (i.e., Lines 25 and 33). We also added the reference suggested (French et al. 2015) about the Phanerozoic (Lines 43-45) to the introduction.

2) Some passages need to be rewritten for clarity, specificity, accuracy. I recommend that the authors revisit the manuscript again before publication to improve the text. I have listed some examples here.

a. Lines 33-34. This vague statement could be revised to a more specific statement such as “modulate production of metabolic markers of phototrophic sulfur bacteria that are used to identify photic zone euxinia in the geologic past”

Response: Rephrased as recommended: “These observations show that viruses have the genomic potential to modulate the production of metabolic markers of phototrophic sulfur bacteria that are used to identify photic zone euxinia in the geologic past.” (Line 32-33).

b. Lines 55-57. This sentence needs rewriting since it suggests that chlorobactene, isorenieratene, and okenone are the diagenetic products, which is inaccurate as these are the biological precursor carotenoid pigments.

Response: Rephrased for clarity: “Based on the ecology of modern euxinic basins, the preservation of diagenetic products of GSB carotenoid pigments (chlorobactene and isorenieratene, preserved as isorenieratane and chlorobactene, respectively) is interpreted as a marker for a deeper photic zone, compared to where PSB pigments (okenone, preserved as okenane) are found.” (Line 56-60).

c. Lines 94-95: suggest something like “phage regulation of phototrophic sulfur bacteria pigment synthesis may affect the abundance and distribution of the GSB and PSB biomarkers that are used as indicators of PZE in the rock record” as it is more specifically worded. Otherwise, aspects of this sentence are vague.

Response: Rephrased as recommended: “For example, phage regulation of phototrophic sulfur bacteria pigment synthesis may affect the abundance and distribution of GSB and PSB biomarkers that are used as indicators of photic zone euxinia in the rock record.” (Lines 96-98).

d. Line 124: revise to “was extracted for DNA” since pigment extraction is also possible and could be relevant to this manuscript.

Response: Rephrased as recommended (Line 398).

e. Lines 249-252: rewrite the sentence to make it more accurate and specific. “Known producers of diagenetic products” would be more accurately written as “Known producers of okenone, which is the biological precursor of the biomarker okenane,...” or “...which is the precursor of the diagenetic product okenane”.

Response: Rephrased as recommended (Line 126-129).

f. Line 389- rewrite since the sedimentary products are not light-harvesting necessarily, rather they are diagenetically related to light harvesting pigments.

Response: Corrected as follows: The diagenetic products of light-harvesting pigments (i.e., chlorobactene, isorenieratene, and okenone) preserved in sediments and in the geologic record are used as proxies of the photic zone euxinia.” (Lines 265-267).

g. Line 392: suggest revising to “...pigments in sediments” since sedimentary record applies more ancient timescales and the cited studies are from modern sediments.

Response: Changed as recommended: “However, previous studies have shown a decoupling between the abundance of GSB and PSB and the concentrations of their pigments in sediments of modern environments.” (Lines 267-269).

h. Lines 451-453: Needs revision. The comma is confusing or a word is missing.

Response: A missing ‘and’ was added (Line 327).

i. Lines 480-482: Rewrite and simplify wording

Response: Rephrased for clarity: “Our observations introduce the potential for applying isotope data to infer viral effects on microbial sulfur cycling.” (Lines 355-357).

j. Lines 504-505: Simplify. Suggest “Our observations suggest that viral infections could impact biosignatures of phototrophic sulfur bacteria in the sedimentary record.”

Response: Rephrased as recommended (Line 377-378).

3) Would it be reasonable to expand on lines 419-420 based on the results? For example, would it be reasonable to suggest that viral infection could lead to higher

okenone abundances such that okenane abundances in the sedimentary record could overestimate PSB abundances and therefore the shallowness of PZE? Would the presented result and results from other systems, such as Green Lake, support this more specific statement?

Response: A statement to this effect has been added: “For instance, viral infection could lead to higher okenone production and consequent okenane preservation in the sediments. This would lead to an overestimation of PSB and, therefore, an inaccurate interpretation of shallow photic zone euxinia.” (Lines 296-299).

Reviewer #3 (Remarks to the Author):

I have enjoyed reading the revised version of this manuscript. It is clear that the authors have gone to some lengths to address my questions regarding the provenance and abundance of the viral AMG_s identified, and the validity of the MAG_s generated. The modifications made to the methods section improves clarity and reproducibility, and the updated version of the discussion better suits the scope of the results and is far easier to follow. I am happy to conclude that the authors have addressed my previous concerns.

In addition, it is clear that the authors have thoroughly addressed Reviewer #4's concerns. The discussion on possible alterations to host sulfur metabolism and the revisions to Figure 4 (now Figure 5) and do an excellent job in clarifying the authors' ideas of how different viral AMG_s could alter metabolic markers used to interpret the geologic record (Reviewer #4's chief concern). I do think this MS is looking very solid now.

We thank the reviewer for the comments.